**Solar forced diurnal regulation of cave drip rates via phreatophyte evapotranspiration**
Katie Coleborn[1], Gabriel C. Rau[1], Mark O. Cuthbert [2], Andy Baker[1], Owen Navarre[3]
[1]*Connected Waters Initiative Research Centre, UNSW Australia, Kensington, NSW 2052*
*Australia*
[2]*School of Geography, Earth and Environmental Sciences, University of Birmingham,*
*Edgbaston, Birmingham, B15 2TT, UK*
[3]*School of Biology Earth and Environmental Sciences, UNSW Australia, Kensington, NSW*
*2052 Australia*
Corresponding author: Katie Coleborn
Email: k.coleborn@unsw.edu.au
Tel: +61434105636
**Abstract**
We present results of a detailed study of drip rate variations at 12 drip discharge sites in
Glory Hole Cave, New South Wales, Australia. Our novel time series analysis, using the
wavelet synchrosqueezed transform, reveals pronounced oscillations at daily and sub-daily
frequencies occurring in 8 out of the 12 monitored sites. These oscillations were not
spatially or temporally homogenous, with different drip sites exhibiting such behaviour at
different times of year in different parts of the cave. We test several hypotheses for the
cause of the oscillations including variations in pressure gradients between karst and cave
due to cave breathing effects or atmospheric and earth tides, variations in hydraulic
conductivity due to changes in viscosity of water with daily temperature oscillations, and
solar driven daily cycles of vegetative (phreatophytic) transpiration. We conclude that the
only hypothesis consistent with the data and hydrologic theory is that daily oscillations are
caused by solar driven pumping by phreatophytic trees which are abundant at the site. The
daily oscillations are not continuous and occur sporadically in short bursts (2-14 days)
throughout the year due to non-linear modification of the solar signal via complex karst
architecture. This is the first indirect observation of tree water use in cave drip water and
has important implications for karst hydrology in regards to developing a new protocol to
determine the relative importance of trends in drip rate, such as diurnal oscillations, and
how these trends change over timescales of weeks to years. This information can be used to
infer karst architecture. This study also demonstrates the importance of vegetation on
recharge dynamics, information that will inform both process-based karst models and
empirical estimation approaches. Our findings support a growing body of research exploring
the impact of trees on speleothem paleoclimate proxies.
**1.      Introduction**
Karst architecture determines the flow and storage of water from the surface to the
underlying cave and is a major influence on drip discharge. Karst systems are characterised
by three principle flow types. Primary flow occurs where the water travels through the
primary porosity of the rock matrix, secondary flow pathways are characterised by water
transported along fractures in the bedrock and tertiary flow pathways consist of conduits
enlarged by dissolution. The dominance of a particular flow regime changes over time, for
example, older limestone tends to have higher secondary porosity (more fractures and
enlarged conduits) and a lower primary porosity due to compaction or cementation (Ford
and Williams, 1994). The relationship between karst architecture and delivery of water to
cave drip discharge sites has been studied to constrain uncertainty in paleoclimate studies
(Bradley et al., 2010; Markowska et al., 2015), identify suitable speleothems as climate
archives (McDonald and Drysdale, 2007) and in conjunction with drip water geochemistry to
determine water residence times in karst aquifers (Arbel et al., 2010; Fairchild et al., 2000;
Lange et al., 2010; Sheffer et al., 2011; Tooth and Fairchild, 2003; Treble et al., 2013b).
Recent research examining drip hydrology and fluctuations in drip rate have used
hydrological response to characterise flow paths. For example,  Markowska et al., (2015)
used statistical analysis of drip hydrology data to identify storage flow, in both the epikarst
and overlying soil, to develop conceptual models of a karst system.
Over a timescale of months to years, fluctuations in drip discharge are typically driven by
seasonal variation in water availability (Hu et al., 2008; Sondag et al., 2003) and long-term
climate forcings such as the North Atlantic Oscillation or El Niño-Southern Oscillation
(McDonald, 2004; Proctor et al., 2000). On a daily to weekly timescale, drip rate responds to
individual rainfall events (Baldini et al., 2012) and barometric changes (Genty and Deflandre,
1998; Jex et al., 2012; Tremaine and Froelich, 2013). Tremaine and Froelich (2013) found
weekly and daily fluctuations at one drip site where an increase in barometric pressure
decreased volumetric drip rate. This was attributed to atmospheric tides, the heating and
cooling of the atmosphere, as the diurnal cycles occurred at two hours before the solar
noon (S1) and solar midnight (S2) each day. The cave was situated in poorly to moderately
indurated Oligocene limestone with a high likelihood of primary porosity (Scott, 2001). Jex
et al. (2012) observed a negative correlation between weekly barometric pressure changes
and drip rate at two out of forty drip sites monitored at the base of a paleokarst feature in
the marmorised and fractured Devonian limestone at Cathedral Cave, NSW. One drip
discharge site had a relatively strong anti-correlation (R=-0.52) after accounting for a 40 hr
time lag. This relationship was attributed to a two-phase flow, where pressure fluctuations
expanded and compressed air bubbles in the water held within the paleokarst in the
unsaturated zone.
Non-linear and chaotic behaviour of drip discharge has been observed over very short
(second to minutes) timescales. Chaotic drip regimes were first noted by Genty and
Deflandre (1998) in the Devonian limestone of southern Belgium (Genty and Deflandre,
1998).  Chaotic and non-linear drip responses were also observed at an event-scale in the
fractured-rock limestone of Cathedral Cave, NSW (Mariethoz et al., 2012). These were
attributed to the filling and draining of subsurface karst stores within a recharge event, with
increasing homogenisation of flow with the filling of the stores. Baker and Brunsdon (2003)
observed non-linear responses to rainfall in multi-year drip time series from a fractured rock
(Carboniferous limestone) in Yorkshire, UK. With the exception of Tremaine and Froeclich
(2013), daily fluctuations have not been observed in cave drip water hydrology. In this paper
we aim to increase our understanding of karst architecture by using a novel approach, the
synchosqueeze transform, to analyse drip discharge time series from 12 drip discharge sites
in Glory Hole Cave, SE Australia. This analysis allows us to characterise daily and sub-daily
fluctuations in drip rate and identify the processes driving these oscillations. This study has
important implications for understanding karst unsaturated flow processes and karstic
groundwater recharge. Currently, most karst models use very simplistic representations of
unsaturated flow, if it is considered at all (Hartmann et al., 2014a). This study highlights the
importance of vegetation dynamics on vadose flow and recharge making it significant to
karst modelling research and speleothem-based paleoclimate studies which focus on the
impact of vegetation dynamics on proxy records (Treble et al., 2015, 2016)

## 2.     Field site and methods

### 2.1 Glory Hole Cave at Yarrangobilly Caves National Park

Glory Hole Cave is part of the Yarrangobilly Caves National Park located in the Snowy Mountains, New South Wales, Australia (35°43'29.3"S 148°29'14.9"E) at an elevation of 980 m (Australian Height Datum). The Snowy Mountains forms part of the Great Dividing Range, a mountainous region stretching along the eastern seaboard from Queensland to Victoria. The region is sub-alpine and the climate is classified as temperate montane with mild summers and no dry season (Köppen climate classification Cfb) (Peel et al., 2007; Stern et al., 2012).

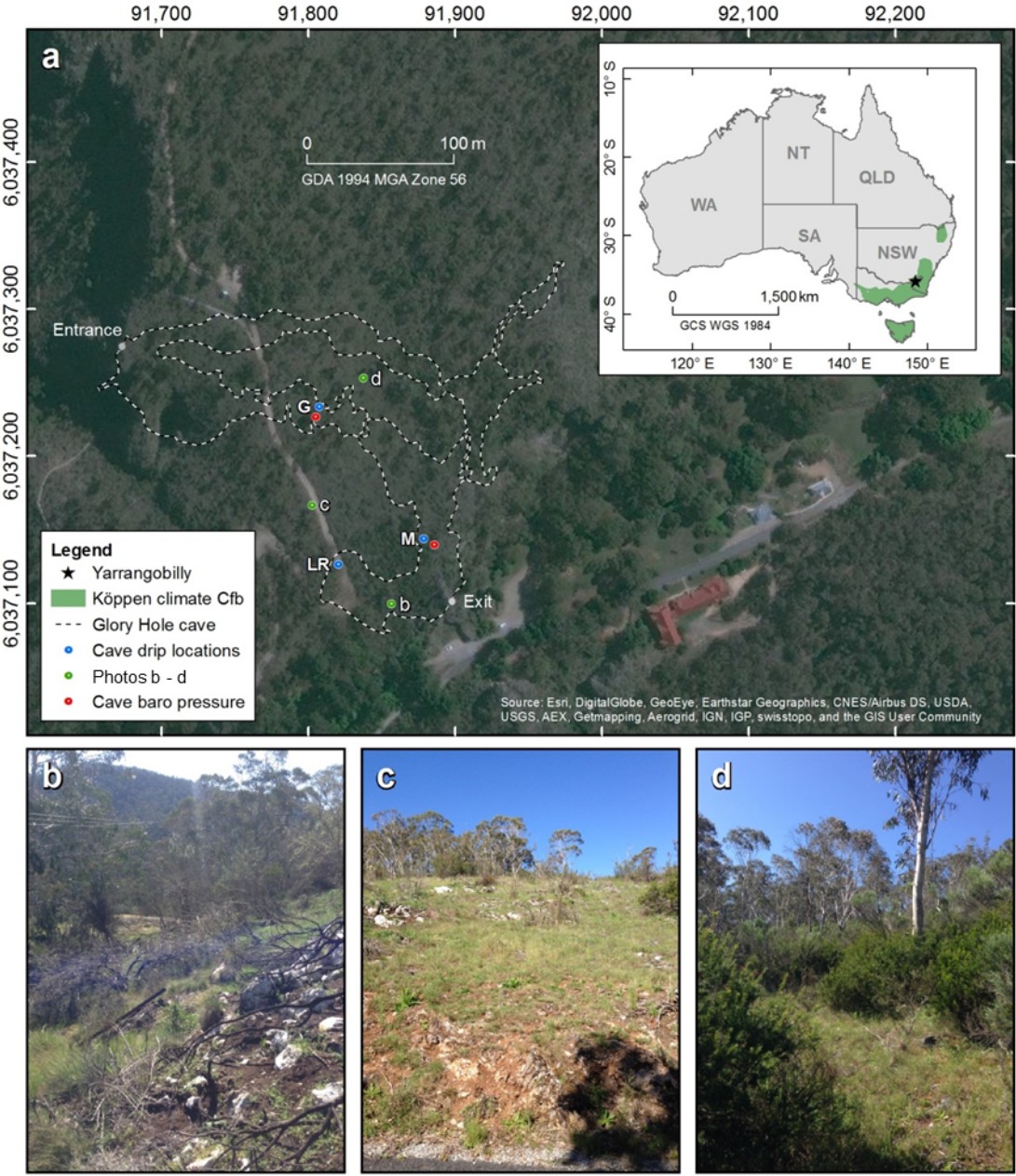

Figure 1 location of Yarrangobilly Caves in New South Wales, Australia with photos of
surface vegetation b-d.  Extent of Köppen climate zone is from Peel et al. (2007).
Glory Hole Cave is formed of two main sections connected by a narrow constriction ~2 m x 6
m. It is ~243 m in length and is ~100 m at its widest point. The cave extends more than 40 m
below the surface in an unsaturated zone of westward sloping limestone bedrock with a
contributing catchment area of ~1 km$^2$. The cave is situated within a formation of massive
limestone approximately 12 km long and on average 1 km wide (Worboys, 1982). The
limestone is typical of south-eastern Australian limestone; it is Silurian, highly fractured and
marblised with little primary porosity. The bedding planes of the limestone are generally
obvious with a westward dip (Adamson and Loudon, 1966). It is likely that Glory Hole Cave
was formed by water running off less permeable rocks to the east of the limestone, sinking
to the water table and rising through large springs close to the Yarrangobilly River (Spate,
2002) which is situated in a gorge in <100 m west of the cave entrance. Glory Hole Cave is
likely to be relevant for paleoclimate proxies as there is an abundance of speleothems and
in close proximity (<100 m) to caves that have been used in multi-proxy speleothem based
paleoclimate studies (Markowska et al., 2015; Webb et al., 2014).
The vegetation is classified as sub-alpine open snowgum (*Eucalyptus pauciflora subsp.*
*pauciflora*) and black sallee (*E. stelullata*) woodland.

## 2.2    Cave and surface monitoring

Drip discharge rate was recorded at 12 drip sites in three locations (Fig. 1 and Table 1)
within Glory Hole Cave using Stalagmate© drip loggers between December 2012 and
September 2015, and monitoring is ongoing. The drip sites were chosen using a stratified
sampling method. A transect of the cave was used to select three locations (G, M and LR)
that satisfied the following criteria 1) there were actively dripping speleothems, 2) spatially
distant from the other locations and 3) different depths within the cave. Individual drips
were sampled randomly at each location, with selection guided by practical constraints such
as stalagmite surface being suitable for placement of logger and the drip falling from high
enough to activate pressure sensor on the logger. Drip loggers recorded the frequency of
drips falling onto the surface of the sealed box containing an acoustic sensor in 15 min
intervals. The number of drips were converted to ml min$^{-1}$, assuming that 1 drip equals 0.19
ml (Collister and Mattey, 2008; Markowska et al., 2015). Recently, automated drip loggers
have been widely used in cave hydrology research (Cuthbert et al., 2014b; Hu et al., 2008;
Mahmud et al., 2015; Rutlidge et al., 2014; Treble et al., 2013a) as they provide a more
convenient and efficient way of recording higher temporal resolution data than traditional
drip counting methods.

Table 1 Summary of drip sites and location within cave as indicated in Fig. 1, the monthly
mean and standard deviation (std) of total flow volume and maximum and minimum drip
rate in summer (December- February) and winter (June- August).

| | | Total flow volume (L) | | | | Drip rate (ml min$^{-1}$) | | | |
|---|---|---|---|---|---|---|---|---|---|
| | | Summer | | Winter | | Summer | | Winter | |
| Site | Location | mean | std | mean | std | Maximum | Minimum | Maximum | Minimum |
| G1 | | 72.67 | 9.21 | 209.58 | 107.78 | 19.51 | 1.84 | 56.75 | 0.00 |
| G3 | | 23.76 | 10.13 | 115.44 | 8.37 | 7.00 | 0.00 | 34.43 | 0.00 |
| G6 | G | 3.73 | 1.90 | 16.45 | 0.10 | 1.43 | 0.10 | 4.10 | 0.65 |
| G8 | | 6.36 | 0.49 | 5.81 | 0.16 | 1.11 | 0.00 | 0.96 | 0.34 |
| G10 | | 32.47 | 23.08 | 104.54 | 73.58 | 9.97 | 0.04 | 27.27 | 0.00 |
| G12 | | 6.57 | 5.71 | 9.74 | 4.39 | 1.68 | 0.00 | 2.04 | 0.43 |
| LR1 | LR | 32.31 | 23.93 | 98.62 | 7.39 | 58.30 | 0.00 | 57.77 | 0.00 |
| M1 | | 0.29 | 0.18 | 0.47 | 0.00 | 0.13 | 0.00 | 0.11 | 0.00 |
| M2 | | 7.67 | 12.85 | 120.09 | 21.21 | 42.53 | 0.00 | 74.30 | 0.00 |
| M4 | M | 0.88 | 1.47 | 33.95 | 5.17 | 4.02 | 0.00 | 28.45 | 0.00 |
| M10 | | 24.53 | 34.68 | 127.79 | 51.36 | 13.95 | 0.00 | 27.56 | 0.00 |
| M13 | | 7.33 | 5.05 | 67.03 | 6.60 | 12.40 | 0.09 | 41.80 | 0.92 |


Barometric pressure and air temperature were recorded at two locations within the cave
(Fig. 1) using Solinst level loggers at 15 min intervals from January-September 2015.
Precipitation (accuracy ± 4% of total mm), wind speed (accuracy ± 0.1 kph), relative
humidity (accuracy ±2%), air temperature (accuracy ± 0.5 °C) and barometric pressure
(accuracy: ±1.0 kPa) were measured with a Davis Vantage Pro 2 weather station <1 km from
Glory Hole Cave at 15 min intervals and data stored using a Datataker DT80 data logger.
Solar radiation (W m$^{-2}$) was derived from satellite imagery processed by the Bureau of
Meteorology from the Geostationary Meteorological Satellite and MTSAT series.
Daily potential evapotranspiration was estimated using ETo Calculator software developed
by the Land and Water Division of the Food and Agriculture Organisation of the United
Nations http://www.fao.org/nr/water/eto.html. The software is based on the Penman-
Monteith equation and is a physically-based method with physiological and aerodynamic
parameters. The climate parameters used were air temperature (mean, maximum and
minimum), relative humidity (mean, maximum and minimum), wind speed and solar
radiation.

**2.3   Spectral analysis of cave drip discharge rates**
A new advance in signal processing was used to analyse the time-frequency content of
measured cave drip discharge rate, temperature and barometric pressure. Here, the
frequencies of interest are 1 cycle per day (cpd) and faster, i.e. diurnal to sub-diurnal.
Daubechies et al (2011) first presented the wavelet synchrosqueezed transform (WSST) as
an empirical mode decomposition like tool for disentangling a amplitude and phase
modulated signal into approximately harmonic components. Thakur et al (2013) adapted
the WSST to discretised data (rather than continuous functions) and developed a MATLAB
Synchrosqueezing Toolbox (available for download:
https://web.math.princeton.edu/~ebrevdo/synsq/) which efficiently implements the WSST
algorithm and offers a log2 frequency resolution (WSST was officially implemented in
MATLAB as of release R2016a). They further tested the robustness properties of WSST and
found that it precisely estimated key signal components, and that it was stable against
errors and noise (Thakur et al., 2013). The WSST combines advantages of the wavelet
transform in regards to frequency resolution with the frequency reallocation method (Auger
and Flandrin, 1995) in order to reduce spectral smearing when mapping out the time-
frequency content of a complicated signal.
The drip discharge rate time series, barometric pressure and air temperature (potential
weather related drivers of drip discharge oscillations) were analysed for time-frequency
content in the following way:
• Application of the WSST functions in MATLAB (version R2016a or later) or the
Synchrosqueezing Toolbox (Thakur et al, 2013) to compute the signal's frequency
content over time. The output is a 2D matrix containing the complex frequency domain
response $\mathcal{F}_{f,t}$ with elements corresponding to discrete frequency and time values (e.g.,
as rows and columns). Here, $f$ is frequency (in $log_2$ resolution) [1/T] and $t$ is discrete
time (sampling resolution) [T].
• Calculation of the component amplitudes according to the standard signal processing
procedure using
$$A_{f,t} = \left| \mathcal{F}_{f,t} \right| = \sqrt{\Im\left( \mathcal{F}_{f,t} \right)^2 + \Re\left( \mathcal{F}_{f,t} \right)^2} \qquad (1)$$
• Normalisation of the component amplitudes using
$$a_{f,t} = \frac{A_{f_{min}<f<f_{max},t}}{max\left( A_{f_{min}<f<f_{max},t} \right)} \qquad (2)$$
In order to highlight the main frequency components of interest (1 and 2 cpd) we chose
$f_{min} = 0.5$ and $f_{max} = 4$ for the normalisation. However, for other applications
different frequency limits could be useful to identify continuous periods of weaker
frequency components in the presence of other, stronger components as well as chaos.
• Visualisation of the normalised amplitude matrices in pseudo-colour plots. Distinct
frequency components (signals with contrasting amplitudes whose frequency does not
significantly change over time) can easily be distinguished from chaos (i.e., lack of
regular oscillations identified as signals with varying amplitude and frequency over
time). Stronger periodic components would yield larger amplitudes and therefore also a
value that is closer to 1 in the respective WSST plots. While this analysis is conducted
manually, it could be automated using criteria for the strength, continuity and stability
of any frequency component of interest.

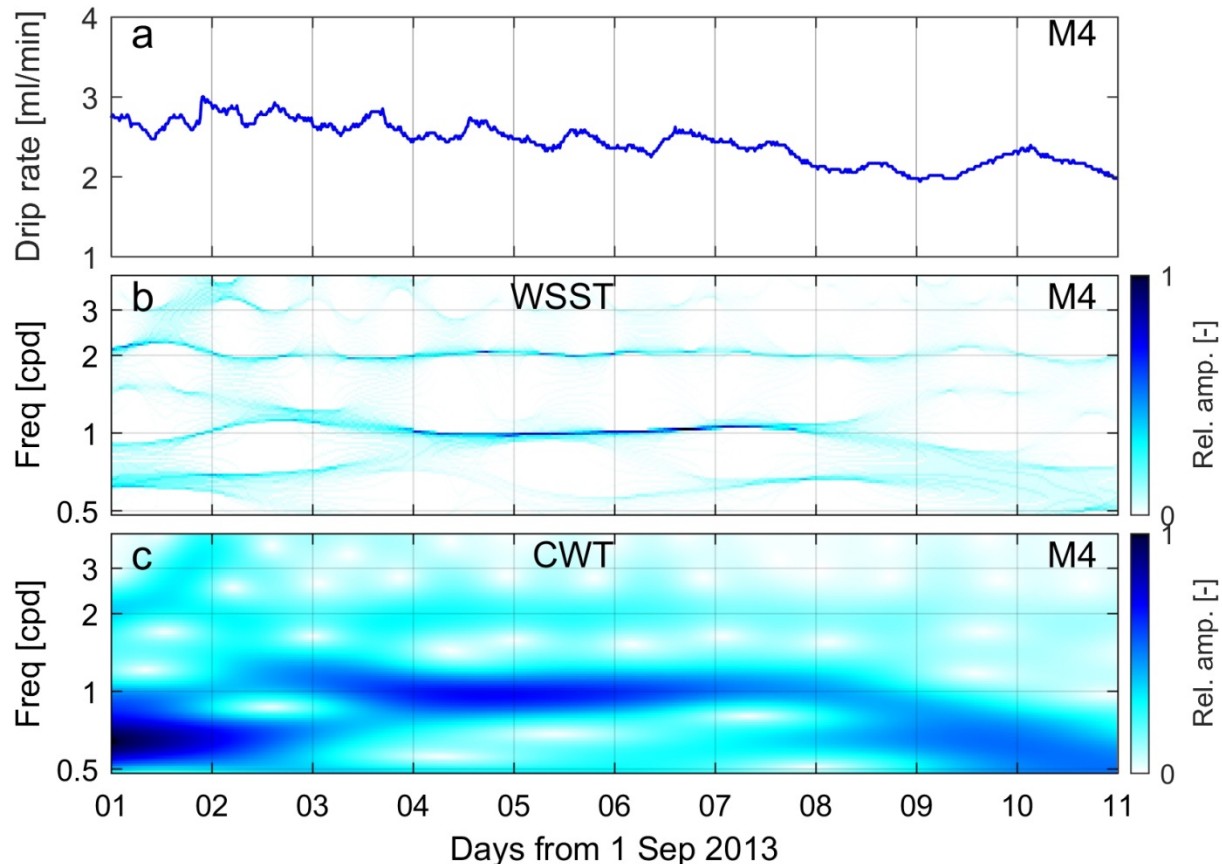


Figure 2 Comparing the time-frequency content of the drip discharge rate within an extract
of data (a) for drip site M4 (refer to Figure 3): Relative component amplitudes calculated
from (b) the wavelet synchrosqueezed transform (WSST), and (c) the continuous wavelet
transform (CWT) using a Morlet mother wavelet.
An example of the time-frequency mapping conducted according to the above described
method is illustrated in Figure 2. Further, the results obtained by applying the WSST (Figure
2b) can be compared to the results from a continuous wavelet transform (CWT) with a
Morlet mother wavelet (Figure 2c) (Torrence and Compo, 1998). From this example it is
clear that WSST features significantly less time-frequency smearing and therefore allows
improved identification and delineation of close-by frequency components such as those at
1 or 2 cpd (compare Figures 2b and 2c). Therefore, WSST presents a significant advantage
over traditional signal processing methods such as the continuous wavelet transform when
identifying the timing and duration of multiple frequency components embedded in
measurements.
Using this methodology, a periodic drip discharge rate could be defined as consisting of
continuous periods of a) stable 1 cpd frequency, b) stable 1 cpd and 2 cpd frequency, c)
chaos (components with randomly varying frequency and amplitude). We used a) and b) as
spectral "fingerprints" to identify and mark periods of continuous occurrence of daily and
sub-daily oscillations in the drip discharge rate dataset.

**Results**

### 3.1 Drip discharge rate time series

The drip discharge time series are presented in Fig. 3. The drip discharge sites are spatially
clustered in three groups within the cave (Fig. 1 and Table 1). Sites with the G prefix are
located near the main entrance of the cave on the western side. The location is highly
decorated with speleothems. M sites are located in the middle section of the cave in a large
chamber with a high ceiling populated by soda straw formations. Location LR1 is situated
near the cave exit at the top of a flow stone.

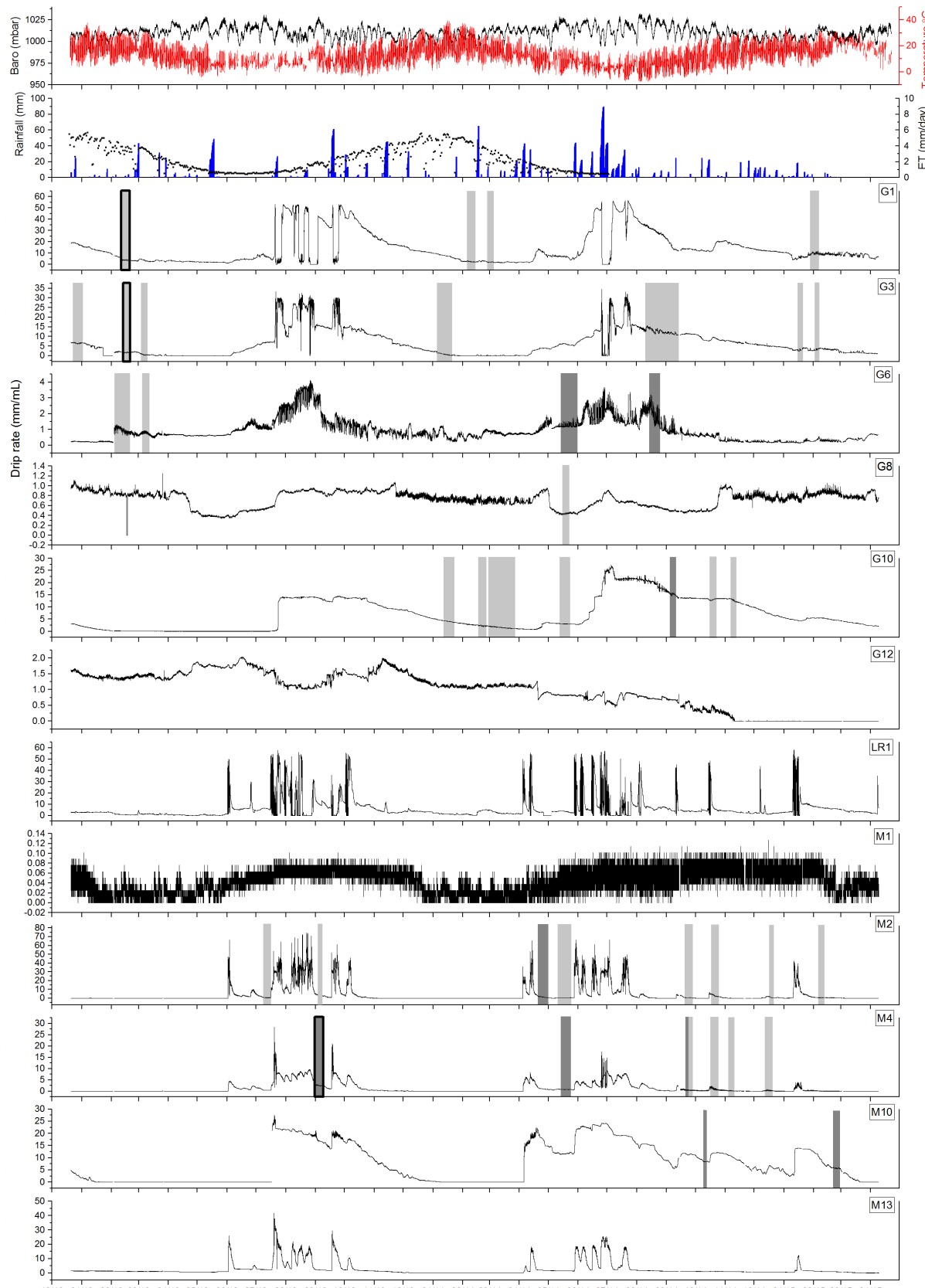



Figure 3 Drip discharge rate time series for all drip sites in Glory Hole Cave with periods
where daily fluctuations occur highlighted in light grey (1 cpd) and dark grey (1 cpd and 2
cpd). The time periods examined in more detail in Fig. 4, 5 and 6 are indicated by bolder
outline. Daily evapotranspiration (19/12/2012- 03/07/2014), rainfall, barometric, air
temperature and are also shown.
The drip discharge rate at G1 and G3 varies seasonally, with higher drip rates in winter, total
flow volume of 133.37 L and 109.52 L, respectively, than summer (64.56 L and 14.1 L,
respectively). Drip rate increases in response to rainfall events during the wet season and
gradually decreases through the drier part of the year.  Drip rate is lowest during April and
May and highest during June and July. Similarly, G6 exhibits seasonal variation with a higher
volume of discharge during the winter than summer. The drip rate at G10 increases sharply
from 0.14 ml min$^{-1}$ on 21/07/2013 to 13.75 ml min$^{-1}$ on 29/07/2013, this drip rate is
consistently sustained for 3 months indicated by the flat topped hydrograph (Fig. 3). From
July 2013 onwards, the drip rate gradually decreases until June 2014 where it increases
sharply again by an order of magnitude from 2.03 ml min$^{-1}$ on 3/06/2014 to 24.96 ml min$^{-1}$
on 4/07/2014. In May 2014, the drip rate again rapidly increases at G10 from 0.142 ml min$^{-1}$
to 21.59 ml min$^{-1}$ on 18/04/2014 and then proceeds to gradually decline until April 2015
where it reaches baseline conditions. M10 exhibits similar behaviour with a low baseline
drip rate which increases sharply during July 2013 and is sustained for ~3 months, however,
the elevated drip rate decreases more rapidly than G10, returning to baseline conditions in
January 2014. M1 has a very low drip rate ranging from 0- 0.13 ml min$^{-1}$ and is seasonally
variable with higher drip rates during the winter. LR1, M2, M4 and M13 are very responsive
to infiltration events and are characterised by a 'flashy' flow type, evidenced by the
frequent spikes in drip rate. G12 has a low discharge rate which gradually decreases over
the monitoring period until the site dries up completely in November 2014. There are small
variations in drip rate that are not associated with rainfall events or seasonal drying. G8 is
the only site which has a lower total flow volume during the winter (2013= 5.92 L; 2014= 5.7
L) than summer (2014= 6.39 L; 2015= 6.84 L).

**3.2 Characterisation of oscillations in the drip discharge rate**
Daily fluctuations in drip discharge rate were identified in eight out of twelve sites using
WSST. There was no connection between the sites that did not exhibit the fluctuations with
respect to spatial location, flow volume or flow regime type. The temporal and spatial
pattern of daily oscillations are shown by the grey shaded areas in Fig. 3. The length of time
the signal is present varied temporally for each drip site. For example, there was a strong 1
cpd signal in the drip water at G1 for 10 days in February 2013 whereas in January 2014 1
cpd fluctuations only lasted 5 days (Fig. 4). The timing of when the signal occurs on an
annual scale varied within and between drip sites. For example, a 1 cpd signal only occurred
during the first 3 months of the year for G1, whereas a 1 cpd signal occurred sporadically at
G3 throughout the calendar year (December 2012, February and March 2013, January 2014,
September 2014, January 2015).
The daily timing of minimum and maximum drip rate varied within and between individual
drip sites. At G1 the 1 cpd maximum and minimum drip rate generally around 6am-12pm
and 12-9pm, respectively. Daily oscillations were only observed once at G8 between 14-
21/05/2014 with minimum drip rate at 3-9 am and maximum drip rate around 12-9pm. Both
1 cpd and 2 cpd signals were observed at M10 for all the periods of drip rate oscillation with
the larger peak occurring in the afternoon around 3-6 pm , minimum drip rate appeared
consistently between 6-9 am. Time lag between drip rate and air temperature was
quantified by performing a cross correlation analysis with a shift interval of 15 mins up to
±24 hours (Table 2). The lag time was identified as the point of maximum negative
correlation between the two variables with the exclusion of sites with missing data. At most
sites the lag time between maximum air temperature and minimum drip rate varied greatly
over the monitoring period. For example, at M4 initially the lag time was 24 hours in
September 2013, decreasing to 9 hours in May 2014 and eventually levelling off at around
16 hours from September to December 2014. In contrast, G1 had a similar lag time over all 4
periods of drip rate fluctuation ranging from 11.25- 12.75 hours. G6 was unique in that the
minimum drip rate occurred before the maximum air temperature in February and March
2013, January 2014 and 2015. Analysis of variance indicated that drip site and season did
not explain a significant amount of variance in lag time.

Table 2 shows the time lag calculated using cross correlation analysis between air
temperature and daily drip rate for each period of drip rate oscillation, the timing of  when
minimum and maximum drip rate occurred within the time periods * denote periods where
a 2cpd signal occurs.

| | Drip rate oscillation period | | | | Max drip rate | | Min drip rate | |
|---|---|---|---|---|---|---|---|---|
| Site | Start | End | Time lag (hours) | $R^2$ (p-value<0.05) | From | To | From | To |
| G1 | 11/02/2013 | 21/02/2013 | -11.5 | -0.82 | 9:00 AM | 12:00 PM | 6:00 PM | 9:00 PM |
| | 4/02/2014 | 14/02/2014 | -12.75 | -0.55 | 9:00 AM | 12:00 PM | 6:00 PM | 9:00 PM |
| | 27/02/2014 | 10/03/2014 | -11.25 | -0.37 | 9:00 AM | 12:00 PM | 12:00 PM | 9:00 PM |
| | 27/01/2015 | 5/02/2015 | -11.5 | -0.69 | 6:00 AM | 12:00 PM | 12:00 PM | 9:00 PM |
| G3 | 23/12/2012 | 2/01/2013 | -23.25 | -0.46 | 12:00 PM | 12:00 AM | 12:00 AM | 9:00 AM |
| | 12/02/2013 | 20/02/2013 | 2 | -0.56 | 3:00 PM | 12:00 AM | 6:00 AM | 3:00 PM |
| | 4/03/2013 | 10/03/2013 | 1 | -0.44 | 3:00 PM | 12:00 AM | 3:00 AM | 9:00 AM |
| | 6/01/2014 | 20/01/2014 | 7 | -0.62 | 12:00 AM | 9:00 AM | 12:00 PM | 6:00 PM |

| | | | | | | | | |
|---|---|---|---|---|---|---|---|---|
| | 20/09/2014 | 29/09/2014 | -4 | -0.38 | 9:00 AM | 6:00 PM | 3:00 AM | 6:00 AM |
| | 16/01/2015 | 20/01/2015 | 0.25 | -0.59 | 6:00 PM | 9:00 PM | 3:00 AM | 9:00 AM |
| | 3/02/2015 | 6/02/2015 | 1 | -0.74 | 3:00 PM | 9:00 PM | 6:00 AM | 9:00 AM |
| G6 | 3/02/2013 | 19/02/2013 | -4 | -0.19 | 6:00 PM | 12:00 AM | 3:00 AM | 6:00 AM |
| | 5/03/2013 | 12/03/2013 | -3.25 | -0.51 | 3:00 PM | 9:00 PM | 3:00 AM | 9:00 AM |
| | 13/05/2014 | 29/05/2014* | -21 | -0.5 | 3:00 PM | 6:00 PM | 3:00 AM | 6:00 AM |
| | 14/08/2014 | 24/08/2014* | -7 | -0.5 | 3:00 PM | 6:00 PM | 9:00 PM | 12:00 AM |
| G8 | 14/05/2014 | 21/05/2014 | -9.5 | -0.55 | 12:00 PM | 9:00 PM | 3:00 AM | 9:00 AM |
| G10 | 5/10/2013 | 16/10/2013* | -24 | -0.4 | 3:00 PM | 6:00 PM | 3:00 AM | 6:00 AM |
| | 5/01/2014 | 22/01/2014 | -0.5 | -0.32 | 3:00 PM | 9:00 PM | 3:00 AM | 6:00 AM |
| | 18/02/2014 | 24/02/2014 | -3 | -0.46 | 3:00 PM | 9:00 PM | 9:00 PM | 12:00 AM |
| | 4/03/2014 | 23/03/2014 | -2.75 | -0.47 | 3:00 PM | 6:00 PM | 3:00 AM | 9:00 AM |
| | 13/05/2014 | 23/05/2014 | -15 | -0.37 | 3:00 PM | 12:00 AM | 12:00 AM | 6:00 AM |
| | 16/10/2014 | 22/10/2014* | -23 | -0.49 | 3:00 PM | 9:00 PM | 6:00 AM | 9:00 AM |
| | 8/11/2014 | 12/11/2014 | -1.5 | -0.59 | 6:00 PM | 9:00 PM | 6:00 AM | 9:00 AM |
| | 5/02/2015 | 25/02/2015 | -0.25 | -0.33 | 3:00 PM | 9:00 PM | 6:00 AM | 9:00 AM |
| M2 | 3/09/2013 | 7/09/2013 | -15.25 | -0.76 | 3:00 PM | 6:00 PM | 6:00 AM | 9:00 AM |
| | 20/04/2014 | 28/04/2014* | -1 | -0.4 | 12:00 AM | 3:00 AM | 6:00 AM | 9:00 AM |
| | 13/05/2014 | 21/05/2014 | -17.75 | -0.6 | 3:00 PM | 6:00 PM | 6:00 AM | 9:00 AM |
| | 20/09/2014 | 28/09/2014 | -23.75 | -0.4 | 3:00 PM | 6:00 PM | 6:00 AM | 9:00 AM |
| | 18/10/2014 | 25/10/2014 | -2 | -0.31 | 3:00 PM | 6:00 PM | 6:00 AM | 9:00 AM |
| | 5/02/2015 | 10/02/2015 | -20.75 | -0.51 | 12:00 AM | 3:00 AM | 12:00 PM | 3:00 PM |
| M4 | 2/09/2013 | 8/09/2013* | -24 | -0.46 | 12:00 PM | 6:00 PM | 6:00 AM | 9:00 AM |
| | 14/05/2014 | 23/05/2014* | -9 | -0.38 | 3:00 PM | 6:00 PM | 6:00 AM | 9:00 AM |
| | 16/10/2014 | 24/10/2014 | -16.25 | -0.65 | 12:00 AM | 12:00 PM | 12:00 PM | 9:00 PM |
| | 4/11/2014 | 13/11/2014 | -16.5 | -0.62 | 9:00 PM | 3:00 AM | 12:00 PM | 9:00 PM |
| | 12/12/2014 | 22/12/2014 | -16.5 | -0.32 | 12:00 AM | 9:00 AM | 6:00 PM | 9:00 PM |
| M10 | 23/12/2012 | 26/12/2012* | -24 | -0.32 | 3:00 PM | 6:00 PM | 9:00 AM | 12:00 PM |
| | 9/10/2014 | 12/10/2014 | -4.75 | -0.46 | 3:00 PM | 6:00 PM | 9:00 AM | 12:00 PM |



1 cpd and 2 cpd signals can occur concurrently, for example, at M4 between 1-9/9/2013
(Fig. 5). This trend, where the 2 cpd is weaker than the 1 cpd is consistent across all sites
where the two signals coincide. The 2 cpd signal can be visually determined in the raw drip
rate data by a second smaller peak). Examples of characteristic WSST plots alongside the
corresponding raw drip rate and surface temperature data will be discussed in greater detail
below. All WSST analyses have been plotted in the SI.

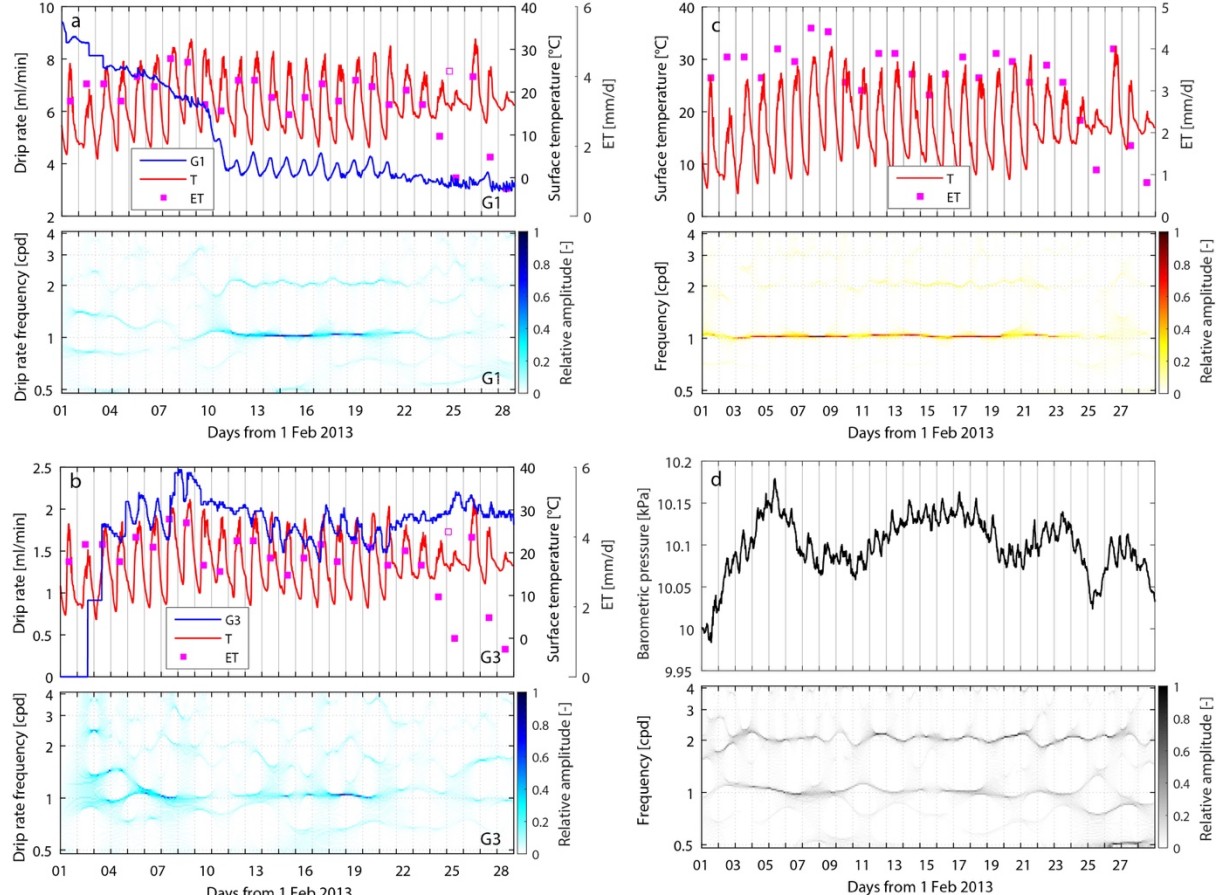


Figure 4 shows the raw drip rate, evapotranspiration and surface temperature data with the
corresponding drip rate WSST plot for time periods where a 1 cpd signal is present for sites
a) G1 and b) G3 c) surface air temperature (T) and potential evapotranspiration (ET) and d)
barometric pressure for period February 2013.


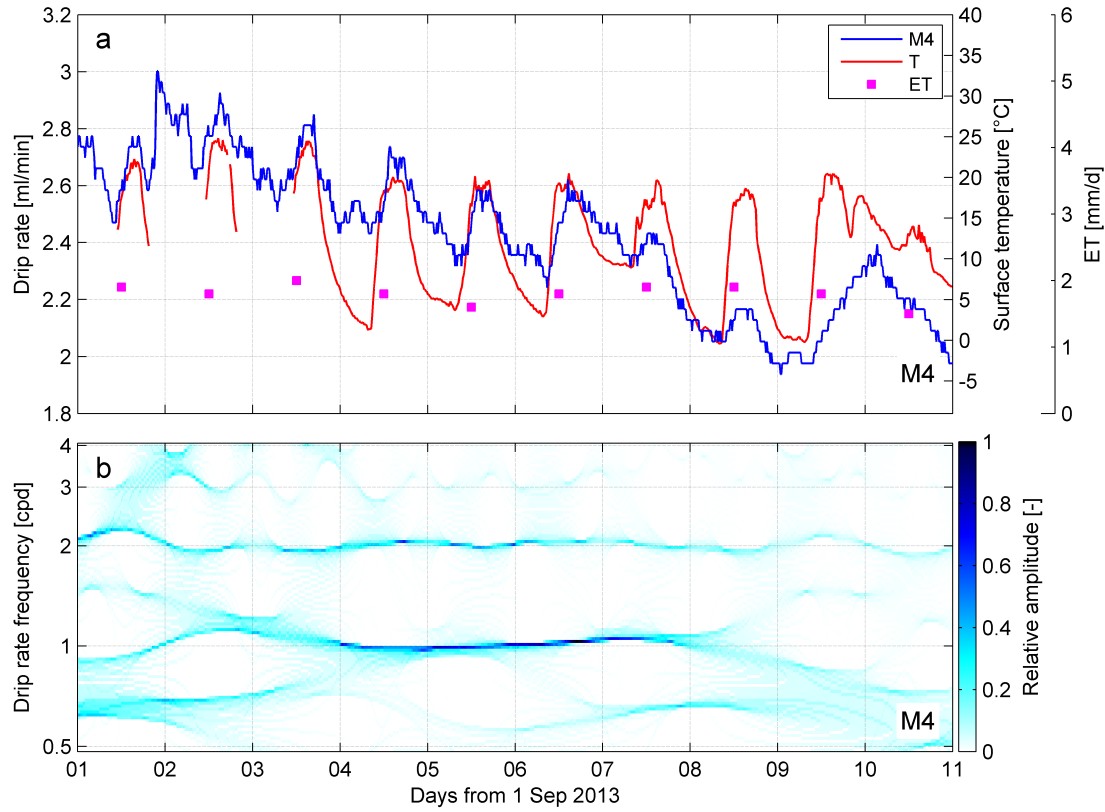


Figure 5 shows the raw drip discharge data, evapotranspiration, surface temperature and
wavelet synchrosqueezed transform (WSST) plot of the drip discharge for site M4 from 1-
11/09/2013.

WSST identified a 1 cpd oscillation in drip rate between 08/02/2013 and 21/02/2013 at G1
and G3 (Fig. 4a, b). At G1 (Fig. 4a), the signal was initially chaotic, but from 08/02/13-
21/02/13 the drip rate oscillates sharply at 1 cpd. The maximum drip rate ranging from 4.03-
3.75 ml min$^{-1}$ occurred between 9:18 and 10:27 and the minimum drip rate ranging from
3.34 -3.75 ml min$^{-1}$ occurred between 18:39 and 21:27. The signal was chaotic from
21/02/2013.

The drip rate at G3 (Fig. 4b) oscillated at 1cpd for 8 days from 12/02/13-20/02/13. In
contrast to G1, the maximum drip rate appeared in the evening and the minimum drip rate
occurred in the morning. The maximum drip rate ranging from 1.63 -2.01 ml min$^{-1}$ occurred
between 20:21 and 00:40 and the minimum drip rate ranging from 0.36-0.48 ml min$^{-1}$
occurred between 9:03 and 11:36 with the exception of 15/02/13 and 18/02/13 where it
appeared at 14:06 and 12:57, respectively. Similar to G1, the 1 cpd trend descended into
chaos from 20/02/13 onwards. The maximum drip rate occurs between 14:23 and 22:45 and
ranged from 0.53 to 1.14 ml min$^{-1}$. The minimum drip rate occurred between 01:18 and
11:32 and ranged from 0.228 to 0.95 ml min$^{-1}$.


From 01-27/02/13, daily barometric pressure peaked between 8:30-9:00 with a magnitude
of 0.1-0.5 kPa with a smaller second peak between 20:00-22:00 with a magnitude of 0.1-0.3
kPa (Fig. 4c). There were larger changes in air pressure on a mesoscale with peaks in air
pressure on 16/02/13, 22/02/13, 26/02/13 and minimum air pressure on 19/02/13,
24/02/13 and 28/02/13. The air pressure changes in these cycles were as much as 1.5-2 kPa.
The drip rate at G1 and G3 did not appear to be affected by the daily or weekly changes in
air pressure. For example, when air pressure decreased dramatically on 27/02/13 (Fig. 4c)
there was no substantial change in drip rate at either G1 or G3.
Insolation drives daily cycles in surface air temperature with maximum temperatures
recorded between 11:30-16:00 and minimum temperatures recorded between 4:00-8:00
(Fig. 4d). The difference in daily minimum and maximum air temperature varied greatly. For
example, between 12- 20/02/2013 the difference was 17.05-22.2 °C whereas between 21-
27/02/2013, the temperature difference was as little as 4.5 °C. Evapotranspiration ranged
from 0.8- 4.5 mm/day and was relatively high from 1-23/02/2013 with a slight downward
trend which then decreased sharply on 23/02/2013 and 24/02/2013 to 2.3 mm/day and 1.1
mm/day, respectively.

## 4. Discussion

### 4.1 Cave drip rate and karst architecture

The complexity of the Glory Hole Cave karst system is evident in the variety of drip regimes. For example, the drip rate at G1, G6 and G3 is seasonally driven with high discharge rates during the wettest period of the year. In contrast, drip discharge at G10 and M10 is likely driven by a storage component which discharges via a less permeable pathway which limits the store at a particular level during wet periods. The drip site is fed via the main water store rather than the overflow pathway itself (Baker et al., 2012; Bradley et al., 2010). Sites LR1, M4, M13 and M2 behave similarly in that they are all very responsive to rainfall events and have low base flows during periods of low rainfall. The response to rainfall events occur within 24 hours across these sites. Calculated flow volumes indicate the storage capacity of the stores feeding the discharge sites. For example, there was an infiltration event on 01/06/2013 which caused a dramatic increase in drip rate for sites LR1, M2, M4 and M13. The flow volumes for each site from the start of the event to the point where the discharge returns to a constant rate are as follows LR1 (1.60 L), M4 (2.99 L), M13 (8.09 L) and M2 (11.30 L). The length of the recession limb, calculated from the peak of the hydrograph until the drip rate returns to base rate, is indicative of the speed at which the store drains. For example, the decay in drip rate is 12 days for site M2 compared to 4 days for M13. The time it takes for the store to drain is not dependent on flow volume, as M13 has a flow volume of more than 5 times that of site LR1 but they both have drainage periods of 5 days. The discrepancy in drainage time could indicate variation in flow pathway length between sites. G8 is the only site with a relatively lower total flow volume during winter than summer. M1 has a low drip rate that shows a small seasonal fluctuation but does not visibly respond to individual events. This site is likely being fed by a store that is large enough to assimilate short term inputs from the surface without impacting drip rate. This type of store has been described as a karst hydrological model component in a number of studies (Arbel et al., 2010; Hartmann et al., 2014b; Markowska et al., 2015).

### 4.2 Daily oscillations in drip rate

Constant frequency oscillations in drip discharge (1 cpd and 2 cpd) occur sporadically throughout the monitoring period December 2012- April 2015 at 8 out of 12 monitored drip sites. This phenomenon could be explained by a number of daily drivers including variations in pressure gradients between karst and cave due to cave ventilation effects, atmospheric and earth tides, or variations in hydraulic conductivity (due to changes in viscosity of water with daily temperature oscillations), and solar driven daily cycles of vegetative (phreatophytic) transpiration. These drivers are now considered in turn.

### 4.2.1. Cave ventilation effects

Surface air pressure and cave air pressure were significantly correlated ($\tau$= 0.82 significant at 95%, n=8939) for the monitoring period 19/01/2015-08/09/2015. This indicates that cave air exchange ("breathing" or ventilation) is very efficient and consequently that variations in air pressure between the cave and surface can be ruled out as driving the fluctuations in drip rate.

### 4.2.2. Barometric loading

Atmospheric tides are caused by changes in air pressure due to the heating and cooling of air masses during the day and night. Correlations between atmospheric tides and drip rates can occur since increases (decreases) in atmospheric pressure at the ground surface are partitioned into stress increase (decrease) in the soil/rock mass and pore pressure increase (decrease) within the formation (Acworth et al., 2015). Drip rates could be affected if this changes the pressure gradient between the groundwater in karst stores and the cave (Tremaine and Froelich, 2013). Such a pressure imbalance is dependent on the hydromechanical properties and karst architecture as well as the degree of pneumatic connection between both the surface and the water table, and the surface and the cave at the location of the drip. Maximum and minimum atmospheric pressure occur at the same time each day (Fig. 4d).

Atmospheric tides were eliminated as a process to explain the daily oscillation phenomenon for several reasons. Firstly, there was no relationship between drip discharge rate and the longer term barometric changes caused by synoptic weather patterns (Fig. 4 ). The mesoscale fluctuations in pressure caused by synoptic weather patterns are an order of magnitude higher than those caused by daily atmospheric tides. Since the drip rate did not respond to pressure changes of this size, they will not respond significantly to changes of a smaller magnitude at a higher frequency because higher frequency signals will be more highly damped and lagged. Secondly, the timing of the daily maximum and minimum drip rates in Glory Hole Cave varied within each drip site over time. For example, the peak discharge time for site G6 varied between 13:24 and 19:48 for the period 11/08/2013-25/08/2015. This finding contrasts with previous studies where drip rate is negatively correlated with barometric pressure and responds to daily pressure changes linearly (Tremaine and Froelich, 2013). However, this could indicate that the daily drip water variations in Glory Hole Cave are being driven by a non-linear process and this is discussed further below. Thirdly, the karst architecture of Glory Hole Cave is well-developed, has little to no primary porosity and is unconfined. Hence, it is unlikely to exhibit barometric responses such as seen in confined systems (Merritt, 2004), whereby pore pressure changes due to barometric loading are substantially lower than the change of cave air pressure.

426

### 4.2.3. Earth tides

Earth tides are solid deformations of the Earth's surface caused by the gravitational pull of the moon and sun (Merritt, 2004). It has been previously shown that earth tides can cause regular oscillations in groundwater level if the aquifer is sufficiently confined (Acworth et al., 2015). However, at Glory Hole Cave this process can be ruled out due to the unconfined conditions, the fact that the compressibility of limestone is smaller than that of water, and because fluctuations in pressure caused by earth tides are so small.

### 4.2.4. Temperature driven viscosity influences on hydraulic conductivity

The study site has large surface temperature variations, particularly in summer where day time and night time temperatures can vary up to 31.1 °C. Consequently, the dynamic viscosity of water could range from 0.8- 1.79 x $10^{-3}$ Pa s (based on a temperature range from 30-0 °C, respectively). However, the conductive propagation in diel temperature variations are expected to be highly attenuated with depth (Rau et al., 2015) resulting in almost complete damping by 1 m bgl. Furthermore, the daily temperature range within the cave itself is just 0.08-1.53 °C, primarily due to air exchange moderated by conductive equilibrium with the cave walls. The variation of water viscosity (which is inversely proportional to hydraulic conductivity) is approximately 2 to 3 % per degree in the range 10 to 30 °C. Considering that the amplitude of a 1 cpd drip rate fluctuation can be as much as 75 % of the maximum drip rate, the greatest anticipated change in hydraulic conductivity, and therefore the drip rate (proportional to the hydraulic conductivity by Darcy's law), on a daily cycle, is likely to be 2-3 orders of magnitude lower than the observed variation in drip rate on a daily basis. We therefore conclude that the daily fluctuations in drip rate are unlikely to be caused by variations in hydraulic conductivity due to changes in viscosity of water.

### 4.2.5. Solar driven daily cycles of vegetative (phreatophytic) evapotranspiration

The timing of the daily drip rate signal appears to be associated with the difference in maximum and minimum surface temperature. In the examples examined in more depth in Fig. 4a-b, when the difference between the maximum and minimum temperature was high (17- 22 °C) and the evapotranspiration was relatively high (mean 3.6 mm/day) the 1cpd signal was continuous. Conversely, when the temperature difference was small (4.5-12.7 °C) and the potential evapotranspiration was relatively lower (mean 2.2 mm/day), the 1 cpd signal dissolved into chaos.

During periods when there are 1 cpd oscillations in drip rate, there was a relationship
between drip rate and surface temperature on a weekly timescale. The best example, in Fig.
6 where $\tau = -0.21$ (significant at 95%) for a 2-day average air temperature and drip rate at G1
from 01-19/02/2014. We have demonstrated above that it cannot be air temperature
driving the signal through either atmospheric tides or water viscosity changes. However, the
relationship between surface temperature variability and 1 cpd drip rate oscillations could
be explained if the association with diurnal temperature variability is due to variations in
solar radiation received at the surface, as it is solar radiation which primarily drives
photosynthesis and thus transpiration in vegetation. This is supported by the fact that solar
radiation is a determining factor in potential evaporation.

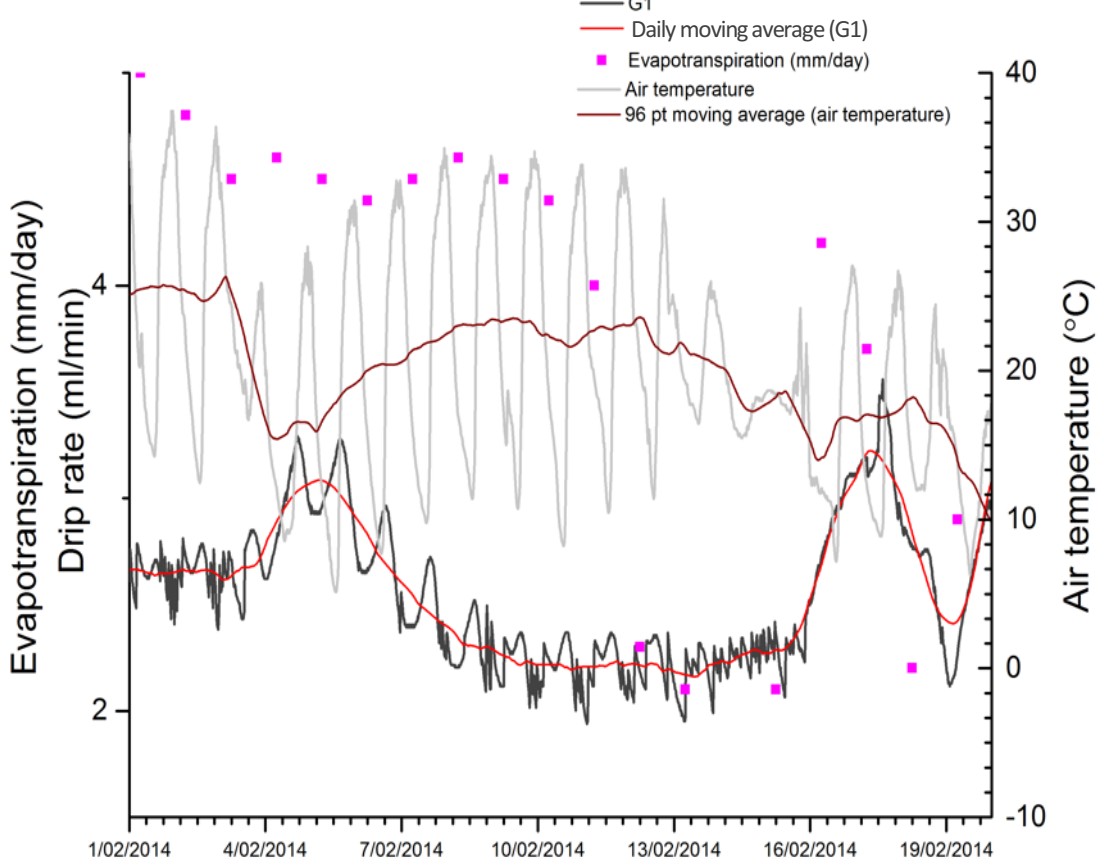


Figure 6 shows the surface air temperature, evapotranspiration and drip discharge rate with
the corresponding daily moving average for site G1 01/02/2014-19/02/2014.

Daytime solar radiation receipt is highest in the absence of cloud cover, because there is no
barrier to incoming short wave radiation which leads to the heating of the earth's surface
and atmosphere, resulting in higher air temperature. Due to the lack of cloud cover, night-
time cooling occurs because of the heat loss through outgoing long wave radiation,
therefore periods of high daytime solar radiation are characterised by large air temperature
amplitudes. In comparison, solar radiation received at the earth's surface is low in the
presence of cloud cover because of the high albedo of clouds. In this case, there is a smaller
temperature amplitude because clouds reduce the amount of incoming short wave
radiation during the day, reducing daytime temperatures and reduce the amount of
outgoing longwave radiation and effectively "insulating" the air at night leading to relatively
warmer temperatures at night. During periods of high solar radiation, plants
photosynthesise more and therefore use more water. We hypothesise that firstly, tree
water use was driving the intermittent daily oscillations in drip discharge demonstrated by
the relationship between daily to weekly variations in surface air temperature and drip
discharge and secondly, the sporadic nature of the oscillations was driven by complexities in
the karst architecture. It has been widely accepted that tree water use causes fluctuations
of the water table (Gribovszki et al 2010; Acworth et al 2015). However, this is the first study
that demonstrates diurnal fluctuation in cave drip rates most likely driven by tree water use.
The area above the cave and in the small uphill catchment is dominated by *E. pauciflora* and
*E. stelullata* (Fig. 1). Eucalypt species have a bimodal root system with shallow lateral roots
and vertically descending roots which penetrate into the profile to depths of up to 18 m,
with depth depending on soil characteristics and the degree to which the bedrock is
fractured and conduits developed  (Crombie, 1992; Farrington et al., 1996). Hence, these
trees have the mechanism to abstract water from karst stores at depth which supports our
theory that tree water use causes daily oscillations in cave drip rate.
Tree water use from deep roots occurs when the upper layers are too dry and have a lower
water potential than the soil water at deeper levels (Dawson and Pate, 1996; Zapater et al.,
2011). Maximum tree water use by the roots is therefore expected in the afternoon during
the period of maximum solar radiation, possibly lagged due to the time taken to
hydraulically lift the water. Conversely, minimum tree water use is expected at the end of
night around 6am. Burgess et al (2001) measured sap flow in Eucalypt tap roots, finding tap
root sap flow peaked around 1 pm and negative sap flow values indicated reverse
(acropetal) flow between 7pm- 7am. In consideration of this, drip water that comes from
fractures and stores which contain tree roots would be expected to have a minimum drip
discharge in the afternoon and maximum around sunrise. In reality, we observe more
complex daily drip oscillations, with peak drip rate occurring at different times of the day
and different times of the year. This is to be expected from a karstified system with flow
routed through a varied and complex fractured network. Different scenarios for daily
oscillations in a karst system will be discussed in detail below.
**4.2.5.1 Scenarios for solar driven daily cycles of phreatophytic evapotranspiration**


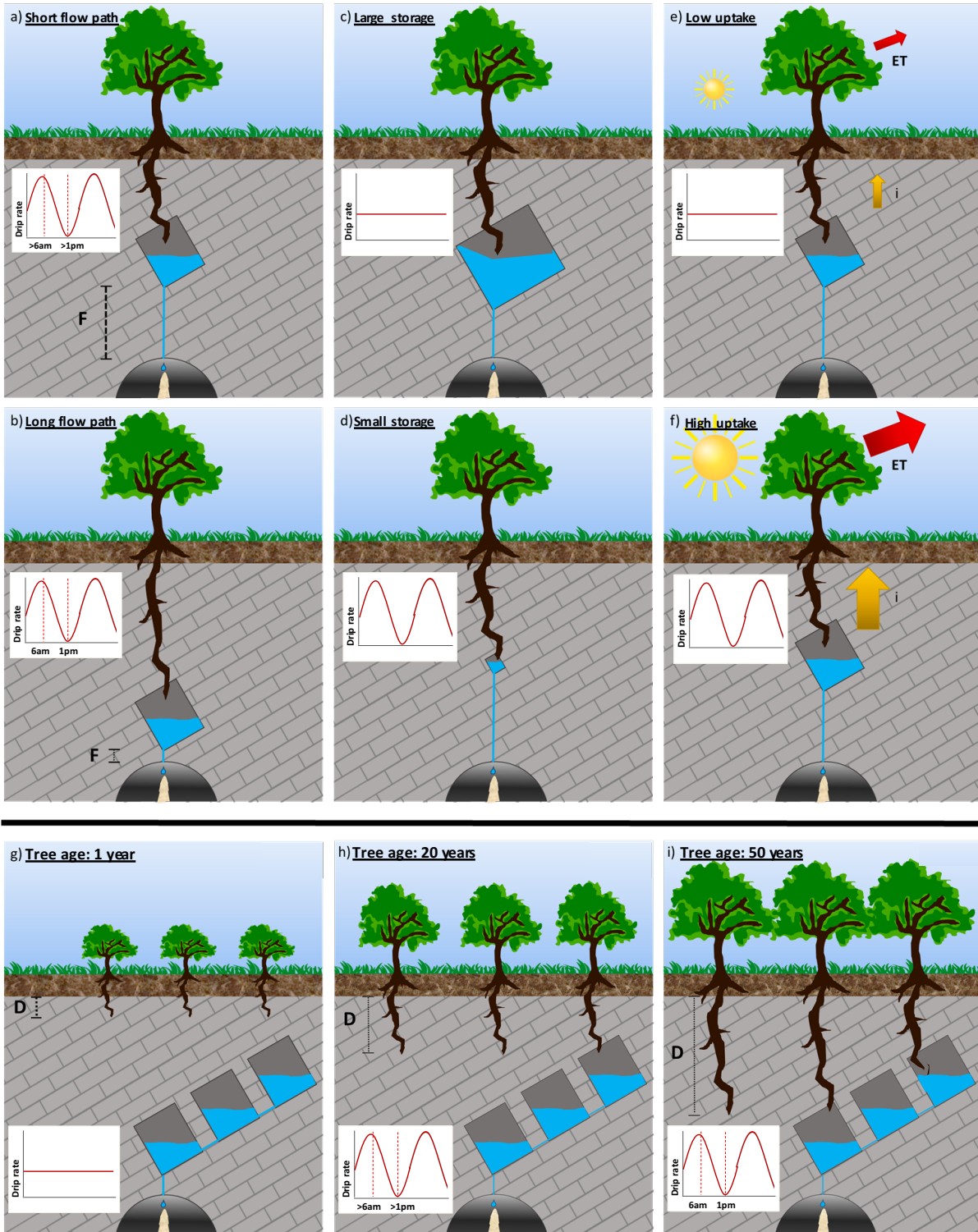


Figure 7 shows a conceptual representation of tree water use from karst stores under different circumstances. a) and b) show different karst store-drip site flow path lengths (F) as the tree roots access karst stores at different depths; c) and d) show tree roots accessing karst stores with different volumes; the influence of annual insolation on evapotranspiration (ET) and root water uptake (i) during winter and summer is shown in e) and f) respectively.

Finally, the increase in rooting depth (L) and access to deeper karst stores over time in years
is explored in g-i.
The depth of a store could affect the timing of daily drip rate oscillations due to the delay in
tree water transport. For example, consider the hypothetical, identical trees with roots
intercepting identical karst stores or fractures at *different* depths in Fig. 7a and 7b. There is
likely to be a greater lag in drip response in Fig. 7a than Fig. 7b because of the longer flow
path-length (F) from the tree root to the cave drip site. Given that eucalypt tap roots can
penetrate to depths ranging from 5-20 m with tap root length depending on the depth of
accessible water (Carbon et al., 1980; Dawson and Pate, 1996) and the drip sites at Glory
Hole Cave are located 30-50 m below the surface, we can speculate that the minimum flow
path length between a taproot accessing the karst store and the drip site below could vary
from 10-45 m. In reality, it is difficult to calculate exact flow path length because of the
prevalence of lateral flow in heavily karstified systems. This has been demonstrated by
Markowska et al (2016) in a study where water spiked with a tracer was used to irrigate the
surface above a cave resulting in a response at discharge sites located 7 m laterally from the
irrigation location. Across all sites, lag time between maximum air temperature and
minimum drip rate ranged from 0.25- 24 hours (Table 2). We can hypothesise that those
sites with a shorter lag time have a shorter path length from tree root accessed store to
cave discharge site than the other drip sites. For example, the lag time for site G1 ranges
from 11.25- 12.75 hours whereas site G10 ranges from 0.5- 3 hours. This process could also
explain the large variation in lag time within a particular site, for example at G6 the lag time
was 21 hours in May 2014 and decreased to 7 hours in August 2014 (Table 2). We
hypothesise that the change could be due to a shortening of the path length from root
accessed store to cave discharge site as the tree grows and increases its rooting depth, thus
accessing a deeper water store. Alternatively, this could also be the result of compensation
as  a shallow water store dries up and a deeper section of the root network is utilised.
The size of the karst store, or volume of water within the store, could determine whether
the daily oscillation is observable or not. Consider the conceptual Fig. 7c and 7d, where
identical trees have roots intercepting different karst stores at the *same* depth. We propose
that a daily oscillation will only be observed when the tree water use is a significant part of
the total water store so a daily oscillation is more likely to be observed in the smaller store
(Fig. 7d) than a store with a  larger volume (Fig. 7c). The influence of store volume on the
presence of daily oscillations could also explain why the phenomenon is not observed at
M1. In section 3.1 we discuss how the low, consistent drip rate at M1 responds to seasonal
drying but does not respond to individual rainfall events. We propose that this site is fed by
a store large enough to assimilate individual rainfall events and the same line of reasoning
could explain the lack of response to tree water use, the volume of water extracted by tree
roots is insignificant in relation to the large volume of water in the store. Conversely, we can
hypothesise that G6 has a small store volume that is more sensitive to water uptake by tree
roots, which is why we see the minimum drip rate occurring 0.25-7 hours before peak air
temperature (Table 2). Furthermore, this scenario is supported by the fact that, generally,
the daily oscillations are not exhibited during periods of high rainfall, and consequently high
drip discharge, as the tree use signal is more likely to be a smaller fraction of the total water
volume. Sites G1, G3 M2 and M4 have high seasonal discharge rates during June-September
as indicated by the multiple hydrograph peaks for the corresponding sites in Fig. 3. There
are no daily oscillations during these periods of peak discharge at any of these sites. Daily
oscillations coincide with the receding limb of the peak at sites M4 (July and September
2013) and M4 (September 2013) as the drip rate decreases. The non-observance of daily
oscillations during periods of high rainfall could also be attributed to the redistribution of
water by the roots from the saturated soil to the unsaturated subsurface (Burgess et al.,
573     2001).

Tree water use responds to annual variation in insolation. Consider Fig. 7e and Fig. 7f where
one tree root intercepts the same karst store over the course of a year. During winter (Fig
7e), there is less insolation than the summer (Fig 7f) therefore the rate of
evapotranspiration is lower. This means that in winter the hydraulic lift (i) is low or negative
and daily oscillations in drip discharge could be dampened or absent. Our analysis reveals
that only 2 out of 41 periods of 1 cpd oscillation occur during winter months June-August
(G6 between 14-24/8/13 and M2 between 8-13/7/2013). However, our analysis also
revealed that season did not explain a significant amount of variance in lag time, thus
suggesting that more variables, such as karst architecture, are affecting the timing of drip
rate oscillations.
In reality, there are multiple trees of different ages above the cave, further complicating the
flow variability. Figure 7g-i presents a conceptual representation of tree tap root length
increasing (L) as the tree grows and accesses deeper karst stores over 1-50 year timescale.
This response to annual insolation and the interaction of multiple trees of varying ages could
explain why daily oscillations at an individual drip site occur one year and not the next, for
example at M10 there is a 1 cpd in December 2012 however, this oscillation does not occur
at the same time in 2013 or 2014. The mechanism in Fig. 7i could also explain why 2 cpd
signals are also observed, whereby multiple tree roots are accessing interconnected water
stores at different depths resulting in two separate cycles with differing lag times. The
occurrence of 2 cpd signals in drip rate could also be related to signal processing where if
the signal is not strictly sinusoidal there may be harmonics in the spectrum. This finding and
the interpretation is an area for further research.
**4.6 Implications for karst architecture and climate proxy modelling research**
Karst architecture controls flow regimes and drip discharge rates of water infiltrating into
caves (e.g., Markowska et al., 2015). Flow rate influences speleothem climate proxies, such
as the $\delta^{18}O$ and concentration of solutes in drip water, through the dilution and mixing of
percolation waters prior to reaching the cave. It is important to distinguish between the
influence of karst architecture and climate-driven processes, such as drought, on discharge
so that paleoclimate proxy records from associated speleothems can be appropriately
constrained. This study has increased our understanding of karst architecture, information
which can be utilised in proxy-system models or forward models, approaches that are
increasingly used to understand cave drip rate variability and to model speleothem proxies
such as $\delta^{18}O$ (Bradley et al., 2010; Cuthbert et al., 2014a). Additionally, we propose that an
important part of any protocol for inferring karst architecture is 1) the incorporation of cave
drip rate monitoring with a minimum 15 min interval at multiple discharge sites for at least a
year and 2) the systematic investigation of daily, weekly and monthly timescales using
frequency analysis capable of showing frequency-time changes, such as the synchrosqueeze
transform (Daubechies et al., 2011) to infer karst flow processes and their relative
importance. This study clearly demonstrates the potential for vegetation to impact karst
water recharge making this research relevant to karst modelling and karst water resources
assessment. Currently, there are no approaches that consider the impacts of vegetation on
recharge dynamics in process-based karst models (Hartmann et al., 2014b, 2015) or in
empirical recharge estimation approaches (Allocca et al., 2014; Andreo et al., 2006).
This is the first indirect volumetric observation of tree water use in cave drip water. This
supports a growing number of studies examining the impact of trees on karst processes and
paleoclimate proxies. For example, tree root respiration provides a source of $CO_2$ for the
dissolution of limestone that is additional to that from soil and vadose zone microbial
respiration. Coleborn et al (2016) found that vegetation regeneration determined post-fire
soil $CO_2$ in a study investigating post-fire impacts on karst processes. Direct observations of
tree water use within the karst unsaturated zone implies the presence of root respiration, a
process which in turn affects drip water and speleothem $^{14}C$ and $\delta^{13}C$ composition (Fairchild
and Baker, 2012; Meyer et al., 2014; Noronha et al., 2015). Trees have been demonstrated
to have long-term effects on cave drip-water solute concentrations. Treble et al. (2015,
submitted) demonstrate long-term trends in drip water calcium and trace element
concentration, which they attribute to increasing solute concentration due to forest
regrowth and increased post-fire tree water use. Baldini et al (2005) infer an effect on
speleothem $\delta^{18}O$ due to secondary forest regrowth after mining and Wong and Banner
(2010) found clearing surface vegetation changed drip water Mg/Ca and Sr/Ca. The findings
and suggested protocol in this study will inform the selection of speleothem specimens for
further research into the impact of tree water use on speleothem paleoclimate proxies.
**5.    Conclusions**

We demonstrated a novel method of analysing recurring patterns in cave water drip rate
using the wavelet synchrosqueezed transform (WSST). Our analysis revealed daily and sub-
daily oscillations with variable temporal and spatial signatures. We tested competing
hypotheses for causes of daily oscillations using drip rate, barometric and temperature data.
The only hypothesis which all the data and hydrologic theory were consistent, was that daily
fluctuations in drip rate were driven by tree water use. We proposed that the complexity of
flow pathways in the karst system accounted for the spatial and temporal variation in the
daily fluctuations of drip rate. This was explored in detail using conceptual models. The
results have wider implications for karst research including providing a new protocol for
inferring karst architecture, informing selection of speleothem specimens for tree water use
paleoclimate studies and highlighting the importance of vegetation dynamics on karst
recharge.

## Author contribution

KC, MOC, GCR and AB wrote the manuscript, discussed the results and implications and
commented on the manuscript at all stages. KC, AB and ON collected data. GCR performed
the WSST analysis and generated the WSST figures. GCR and ON created the location map.
KC generated other graphs and conceptual figures.

## Acknowledgements

We acknowledge that Katie Coleborn was supported the Australian Research Council
(LP130100177). Mark Cuthbert was supported by Marie Curie Research Fellowship funding
from the European Community's Seventh Framework Programme [FP7/2007-2013] under
grant agreement n.299091. We would also like to thank Stuart Hankin for allowing us access
to the weather station data and the National Parks and Wildlife Service staff at Yarrangobilly
Caves. Solar exposure data derived from satellite imagery processed by the Bureau of
Meteorology from the Geostationary Meteorological Satellite and MTSAT series operated by
Japan Meteorological Agency and from GOES-9 operated by the National Oceanographic &
Atmospheric Administration (NOAA) for the Japan Meteorological Agency. We would also
like to acknowledge the use of equipment funded by the Australian Government National
Collaborative Research Infrastructure Strategy (NCRIS).

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
