# Peer review of "Solar forced diurnal regulation of cave drip rates via phreatophyte evapotranspiration"

_Hydrology and Earth System Sciences, 2016_

## Referee Comment (RC1) · Anonymous Referee #1 · 18 Mar 2016

Summary

The manuscript presents data of drip counts in a cave in Australia. The data are analysed with a recent signal processing tool which allows to identify the strength and frequency of periodic signals in a time series. A specific feature of the method is that is allows to identify consecutive periods of the time series which show an periodic signal. This is important for this dataset of cave drip water because there are only a few days within almost two years of measurement which show periodic / diurnal signals. However, the signals are not consistent in space that means they are different at other locations in the cave and they are not consistent in time, that means they do no occur at similar periods. Furthermore the phase of the signals is also not consistent. These spurious occurrences of the period signal may render the finding of a period signal as less important. Still its diagnosis is one of the most important and direct results of this

manuscript.

For the rest of the manuscript the authors try to argue about the origin of the periodic signal. They discuss several earlier proposed causes of the diurnal signal and argue that only a root water uptake could be a reasonable cause. However, there is no direct evidence being presented to undermine this discussion. Therefore I recommend to be much more careful in the wording, e.g. L536 "this is the first volumetric observation of tree water use in cave drip water". I have found a number of other issues, see below, which need clarification. Nevertheless, I think that these issues can be resolved within a thorough revision of the manuscript.

Major remarks

- how representative is the drip measurement? The data shown in Fig. 2 seems to be rather variable and site dependent.

- In the methods section radiation data is being mentioned, but is not used!

- abstract L31: unclear what is meant with "trends in drip rate at different timescales"

- section 2.1: - Is the cave relevant for paleoclimatic proxies? - What is the approximate contributing area to the cave?

- Methods 2.2 - why do you estimate daily potential ET when the focus is set to diurnal variations?

- section 2.3 spectral analysis - clearly describe input and outputs - what is the form of the periodic signal, is it sinusoidal? - By which criteria did you determine the presence of a periodic signal?

Figure 2: - time resolution of drip rates - unit of drip rates

Figure 3: - Y-axis labels on left panels are hidden - for the SST panels it is unclear which time series is transformed? - how is the presence of a significant periodic cycle determined from these plots?

[Figure]

section 4.2.1: the p-value of the t-test is very low suggesting a very low probability of the Null hypothesis of no difference. Thus there is a significant difference of pressure in cave and outside. Anyway I doubt if a t-test on the central tendency is the right tool to assess the ventilation effect. Please check this and revise accordingly.

section 4.2.5: the authors mix up long wave radiative exchange processes and L403-413 need to be revised.

L433: To my understanding deep root water uptake is only required when the upper soil layers get too dry and have a lower potential than the soil water at deeper levels. See papers discussing hydraulic lift (e.g. Dawson, 1996 Tree Physiology, Zapater et al., 2011, Trees). Therefore I think that in the wetter periods no relevant deep root water uptake occurs.

Minor remarks:

- use SI units (L125-L132)

- L272 wrong reference - it should be Fig.3d

- L300f how are recession times being computed?

- L342: there is no Fig 4c

- L439: What is meant with negative hydraulic lift?

---

## Referee Comment (RC2) · Anonymous Referee #2 · 21 Mar 2016

In the submitted manuscript Coleborn at al present a study that deals with the identification and characterization of daily fluctuations of cave drip rates in a karstic cave in New South Wales, Australia. They installed drip counters at 12 locations within the cave and use a method called Synchrosqueezeing to identify periods with stable signals of 1 and 2 drip rate cycles per day (cpds). Such periods could be identified for a subset of the 12 drips, with varying length and signal type (1 or 2 cpds). Comparing the daily signal of those drips with explanatory variables such as air pressure differences between the cave and the atmosphere, the barometric loading due to the daily heating and cooling of air masses, earth tides due to the gravitational influence of the moon, temperature's influence on water viscosity, and solar driven cycles of evapotranspiration activity of the plant cover, they show that evapotranspiration is the most likely reason for daily fluctuations in drip rates. Based on this finding they develop a conceptual model of the impact of vegetation on drip rates under different climatic and structural setups.

Generally this is a very valuable study. It reveals understanding of processes that have not been investigated before. The manuscript is well written and the results are plausible and of high relevance. However, there are some important revisions to be done before publication. My major point of criticism is the lack of quantification of the relations between diurnal fluctuations of drip rates and their explanatory variables. Some few $r^2$s and p-values are provided but the most important part of the discussion (4.2.5. Solar driven daily cycles of vegetative (phreatophytic) evapotranspiration) could definitely use some more quantification of the identified relationships and their significance.

Some specific comments:

1. The introduction needs some information of the relevance of this type of research.

2. The spectral analysis explained in too little detail (schematic figure could be helpful)

3. It is not clear whether the selection of periods of stable cycle per days was based on a threshold procedure ore done manually and subjectively.

4. Implications for karst recharge assessment are missing in the discussion.

Please see the attached commented version of the manuscript for further details.

Please also note the supplement to this comment:
http://www.hydrol-earth-syst-sci-discuss.net/hess-2016-11/hess-2016-11-RC2-supplement.pdf

————————————

[Figure]

**Supplement:**

[revised manuscript text omitted]

---

## Author Comment (AC1) · 20 Apr 2016

Dear Dr Blume,

On behalf of the authors I would like to thank the two anonymous reviewers for their time and efforts in providing detailed and thoughtful feedback that I am confident will greatly improve the manuscript.

As requested we have provided a detailed response to each comment and query raised by the reviewers. We will provide a revised version of the manuscript should our response be deemed competent. Please do not hesitate to contact me if you require further information.

Kind regards,

Katie

**Anonymous Reviewer #1**

The dominance of a particular flow regime changes over time, for example, older limestone
tends to have higher secondary porosity (more fractures and enlarged conduits) and a lower
primary porosity due to compaction or cementation (Ford and Williams, 1994).

The manuscript presents data of drip counts in a cave in Australia. The data are analysed
with a recent signal processing tool which allows to identify the strength and frequency of
periodic signals in a time series. A specific feature of the method is that is allows to identify
consecutive periods of the time series which show a periodic signal. This is important for
this dataset of cave drip water because there are only a few days within almost two years of
measurement which show periodic / diurnal signals. However, the signals are not consistent
in space that means they are different at other locations in the cave and they are not
consistent in time, that means they do no occur at similar periods. Furthermore the phase of
the signals is also not consistent. These spurious occurrences of the period signal may
render the finding of a period signal as less important. Still its diagnosis is one of the most
important and direct results of this manuscript. For the rest of the manuscript the authors
try to argue about the origin of the periodic signal. They discuss several earlier proposed
causes of the diurnal signal and argue that only a root water uptake could be a reasonable
cause. However, there is no direct evidence being presented to undermine this discussion.
Therefore I recommend to be much more careful in the wording, e.g. L536 "this is the first
volumetric observation of tree water use in cave drip water". I have found a number of
other issues, see below, which need clarification. Nevertheless, I think that these issues can
be resolved within a thorough revision of the manuscript.

**Major remarks**

- How representative is the drip measurement? The data shown in Fig. 2 seems to be rather
variable and site dependent.

Thank you for this comment. The drip sites were chosen using a stratified sampling method.
A transect of the cave was used to select three locations (G, M and LR) that satisfied the
following criteria 1) there were actively dripping speleothems, 2) spatially distant from the
other locations and 3) different depths within the cave. Individual drips were sampled
randomly at each location, with selection guided by practical constraints such as the
stalagmite surface being suitable for placement of a logger and the drip falling from high
enough to activate pressure sensor on logger etc. We will include a more detailed site
description to address this issue.

- In the methods section radiation data is being mentioned, but is not used!

We thank the reviewer for this comment. The radiation data was used to calculate daily
evapotranspiration as explained in lines 137-140:

"The climate parameters used were air temperature (mean, maximum and minimum),
relative humidity (mean, maximum and minimum), wind speed and solar radiation."

- Abstract L31: unclear what is meant with "trends in drip rate at different timescales"

Thank you for raising this point. We will reword this phrase to make it clearer. This is the
first observation of tree water use in cave drip water and has important implications for
karst hydrology in regards to developing a new protocol to determine the relative
importance of trends in drip rate, such as diurnal oscillations, and how these trends change
over timescales of weeks to years. This information can then be used to infer karst
architecture.

- Section 2.1: - Is the cave relevant for paleoclimatic proxies? - What is the approximate
contributing area to the cave?

Glory Hole Cave is likely to be relevant for paleoclimate proxies as it is well decorated and in
close proximity (<100 m) to caves that have been used in multi-proxy speleothem based
paleoclimate studies (Webb et al., 2014; Markowska, 2015). The contributing catchment
area is ~1 km$^2$. We will add these details to Section 2.1.

- Methods 2.2 - why do you estimate daily potential ET when the focus is set to diurnal
variations?

Thank you for your queries. The daily potential ET was estimated to show the multi-day
relationship between the presence of drip cycles and ET, rather than just relying on air
temperature. We would like to refer the reviewer to lines 282-285, lines 388-392 and lines
398-404.

- Section 2.3 spectral analysis - clearly describe input and outputs - what is the form of the
periodic signal, is it sinusoidal? - By which criteria did you determine the presence of a
periodic signal?

Thanks for your queries. We have revised the section describing the synchrosqueezing (SST)
procedure in the methodology and it is now much clearer in regards to input and output.
The signal does not have to be sinusoidal. If it is, the frequency content will be much
stronger. Otherwise, it will be decomposed into signal components and spread across
frequency bins. We visually identified the presence of periodic signals from the SST plots, as
is clearly described in the manuscript. We clarified many formulations in this section.
Hopefully, this has answered the questions.

-Figure 2: - time resolution of drip rates - unit of drip rates

We will change the unit of drip rate in Fig 2, thank you for raising this point.

-Figure 3: - Y-axis labels on left panels are hidden - for the SST panels it is unclear which time
series is transformed? - how is the presence of a significant periodic cycle determined from
these plots?

Y-axis labels have now been redrawn in a new Figure 3. For SST panels, we have clearly
marked the time series in the legend as well as inside the respective panels (labels
correspond to the ones in Table 1 and Figure 2).  A response to the last question was given
previously.

-Section 4.2.1: the p-value of the t-test is very low suggesting a very low probability of the
Null hypothesis of no difference. Thus there is a significant difference of pressure in cave
and outside. Anyway I doubt if a t-test on the central tendency is the right tool to assess the
ventilation effect. Please check this and revise accordingly.

We thank the reviewer for this comment. We agree that a t-test was not the best statistical
approach and have instead used Kendall's Tau which shows a strong correlation (0.82)
significant at 0.05.

-Section 4.2.5: the authors mix up long wave radiative exchange processes and L403-413
need to be revised.

We thank the reviewer for bringing this to our attention and will revise the text primarily by
removing 'long wave' as a source of incoming radiation.

-L433: To my understanding deep root water uptake is only required when the upper soil
layers get too dry and have a lower potential than the soil water at deeper levels. See
papers discussing hydraulic lift (e.g. Dawson, 1996 Tree Physiology, Zapater et al., 2011,
Trees). Therefore I think that in the wetter periods no relevant deep root water uptake
occurs.

We thank the reviewer for this information and the suggested references. We agree that
deep root water uptake is only required during drought periods, when the shallow soil is
dry. We will revise the text to reflect the information in the references provided.

Minor remarks:

- use SI units (L125-L132)

We will correct this so that all presented data is in SI units.

- L272 wrong reference - it should be Fig.3d

Thank you for this comment, this reference will be corrected.

- L300f how are recession times being computed?

The recession times are calculated from the peak of the hydrograph to the point when the
drip rate returns to the baseline value. We will clarify this in the text accordingly.

- L342: there is no Fig 4c

Thank you for identifying this mistake, we will correct the reference to Fig 3c.

- L439: What is meant with negative hydraulic lift?

We thank the reviewer for this comment. Negative hydraulic lift refers to the process when
the transpiration rate is low and water is transported from the roots back into the soil. We
will clarify this sentence along the lines of "Burgess et al (2001) measured sap flow in
Eucalypt tap roots, finding hydraulic lift peaked around 1 pm and negative sap flow values
indicated reverse (acropetal) flow between 7pm- 7am."

**Anonymous reviewer #2**

In the submitted manuscript Coleborn at al present a study that deals with the identification and characterization of daily fluctuations of cave drip rates in a karstic cave in New South Wales, Australia. They installed drip counters at 12 locations within the cave and use a method called Synchrosqueezeing to identify periods with stable signals of 1 and 2 drip rate cycles per day (cpds). Such periods could be identified for a subset of the 12 drips, with varying length and signal type (1 or 2 cpds). Comparing the daily signal of those drips with explanatory variables such as air pressure differences between the cave and the atmosphere, the barometric loading due to the daily heating and cooling of air masses, earth tides due to the gravitational influence of the moon, temperature's influence on water viscosity, and solar driven cycles of evapotranspiration activity of the plant cover, they show that evapotranspiration is the most likely reason for daily fluctuations in drip rates. Based on this finding they develop a conceptual model of the impact of vegetation on drip rates under different climatic and structural setups. Generally this is a very valuable study. It reveals understanding of processes that have not been investigated before. The manuscript is well written and the results are plausible and of high relevance.

We thank the reviewer for recognising the value of this manuscript.

However, there are some important revisions to be done before publication. My major point of criticism is the lack of quantification of the relations between diurnal fluctuations of drip rates and their explanatory variables. Some few r2s and p-values are provided but the most important part of the discussion (4.2.5. Solar driven daily cycles of vegetative (phreatophytic) evapotranspiration) could definitely use some more quantification of the identified relationships and their significance.

Thank you for these comments; we will address your concerns point by point below.

Some specific comments:

1. The introduction needs some information of the relevance of this type of research.

We thank the reviewer for bringing this to our attention. We agree that this study has important implications for understanding karst unsaturated flow processes and karstic groundwater recharge. Currently, most karst models use very simplistic representations of unsaturated flow, if it is considered at all (Hartmann et al., 2014a). This study highlights the importance of vegetation dynamics on vadose flow and recharge making it significant to karst modelling research and speleothem-based paleoclimate studies which focus on the impact of vegetation dynamics on proxy records (Treble et al., 2015, accepted for
publication 8/4/16).

2. The spectral analysis explained in too little detail (schematic figure could be helpful)

This issue was also raised by reviewer #1, and we have expanded the description in
response. The synchrosqueezing transform methodology was developed and tested by
other reviewers, and going into detail exceeds the framework of this manuscript. We have
referenced all works required by the interested reader to familiarise with SST.

3. It is not clear whether the selection of periods of stable cycle per days was based on a
threshold procedure ore done manually and subjectively.

We thank reviewer #2 for their comment which was also raised by reviewer #1. We visually
identified the presence of periodic signals from the SST plots, as is clearly described in the
manuscript. We clarified many formulations in this section. Hopefully, this has answered the
questions.

4. Implications for karst recharge assessment are missing in the discussion.

We thank the reviewer for recognising the wider implications of our research and for
providing these excellent references. This study clearly demonstrates the potential for
vegetation to impact karst water recharge making this research relevant to karst modelling
and karst water resources assessment. Currently, there are no approaches that consider the
impacts of vegetation on recharge dynamics in process-based karst models (Hartmann et al.,
2014b, 2015) or in empirical recharge estimation approaches (Allocca et al., 2014; Andreo et
al., 2006).

Please see the attached commented version of the manuscript for further details.

Comments within text

-Not necessarily → please check and cite standard Karst literature as

Goldscheider, N., Drew, D., 2007. Methods in Karst Hydrogeology. Taylor & Francis Group,
Leiden, NL.

Ford, D.C., Williams, P.W., 2013. Karst Hydrogeology and Geomorphology. John Wiley &
Sons.

We thank the reviewer for this suggestion. After consulting Ford and Williams (1994) we will
reword the sentence "the dominance of a particular flow regime changes over time, for
example, older limestone tends to have higher secondary porosity (more fractures and
enlarged conduits) and a lower primary porosity due to compaction or cementation (Ford
and Williams, 1994).

- also worth citing:

Arbel, Y., Greenbaum, N., Lange, J., Inbar, M., 2010. Infiltration processes and flow rates in
developed karst vadose zone using tracers in cave drips. EarthSurf. Process. Landforms 35,
1682–1693. doi:10.1002/esp.2010

Lange, J., Arbel, Y., Grodek, T., Greenbaum, N., 2010. Water percolation process studies in a
Mediterranean karst area. Hydrol. Process. 24, 1866–1879.

Sheffer, N.A., Cohen, M., Morin, E., Grodek, T., Gimburg, A., Magal, E., Gvirtzman, H., Nied,
M., Isele, D., Frumkin, A., 2011. Integrated cave drip monitoring for epikarst recharge
estimation in a dry Mediterranean area, Sif Cave, Israel. Hydrol. Process. 25, 2837–2845.
doi:10.1002/hyp.8046

We thank the reviewer for suggesting these references and agree that they are relevant to
the manuscript. We will include them in the manuscript citations.

-Please add short paragraph on the relevance of such investigations in terms of
understanding Karst unsaturated flow processes and karstic groundwater recharge,
paleoclimate reconstructions etc. Most karst models use very strong simplifxations of
unsaturated karst flow processes, if they consider unsaturated flow at all.

Hartmann, A., Goldscheider, N., Wagener, T., Lange, J., Weiler, M., 2014. Karst water
resources in a changing world: Review of hydrological modelling approaches. Rev. Geophys.
52, 218–242. doi:10.1002/2013rg000443

We thank the reviewer for this comment and for providing a suggested citation. We would
like to direct their attention to our response to 'Specific Comment #2' above.

-don't want to be picky but the meaning of "AHD" might not be obvious to everybody.

We thank the reviewer for this comment; we will expand the acronym to "Australian Height
Datum" for clarity.

-Is this true for all types of drips? Is there some uncertainty involved. Please provide shortly
Some more detail why this exact volume per drop ol water is valid.

Thank you for this comment, the cited paper is an experimental study where the effect of
spherical stalactite diameter and flow rate on drop volume. There is uncertainty when
estimating drop size because the volume changes over time because of variation in flow rate
and the changing morphology of the stalactite. For example, the diameter of the tube in a
soda straw formation could shrink during a period of low flow rate leading to a smaller drop
volume. It is also impossible to quantify the error of these estimates without measuring
physical volumes. We argue that the drop volume 0.19 ml is a reasonable assumption as this
value was used in a study conducted at the same site in a cave with a similar distribution of
stalactite radii and formation (Markowska et al. 2015). We will include a more detailed
explanation in the amended manuscript.

-So is it Potential evaporation? Please clarity.

It is potential evaporation; we will clarify this in the text. Thank you for bringing this point to
our attention.

-The elaborations of this method are not detailed enough. Also there is too much
referencing of secondary literature. Please add some more elaborations, desirably a small
sketch that show how the method works schematically.

This issue was also raised by reviewer #1, and we have expanded the description in
response. The synchrosqueezing transform methodology was developed and tested by
other authors working in the field of signal processing. Going into detail here exceeds the
framework of this manuscript. We recognise that SST is a brand new technique, and that
many readers would be unfamiliar with it. We have therefore clearly referenced all works so
that interested readers can familiarise with this spectral analysis technique.

-Was there a systematic threshold or were stable cpds selected manually. A systematic
selection would be desirable.

We used manual visual identification the presence of periodic signals from the SST plots, as
is clearly described in the manuscript (lines 160-162).

-Please set y-axis to 0, With the present soaking it is difficult identifying when the drips
actually fall dry.

Thank you for this suggestion, we will make the changes to Fig 2 so that y-axis starts at 0.

- Correlation?

We thank the reviewer for this suggestion. We chose the phrase 'connection' rather than
'correlation' because the statement is not based on a statistical outcome.

-Less permeable ?

Thank you for this suggestion, we will make the suggested change in the text.

-Other studies also show this type of behavior. Please see for example the three studies on
cave drip investigations in Israel that I mentioned in a previous comment.

We thank the reviewer for this suggestion, we agree that these references are relevant and
we will add the citations in the text.

-What about hydraulic connectivity? When percolating waters passed the regions where
erapotranspiration takes place evaporation won't affect its flow percolation rate any more,
right?

We thank the reviewer for this comment. If the unsaturated zone above the cave had a
homogenous hydraulic connectivity, we would expect there to be no spatial or temporal
variability in the occurrence of the daily oscillations. However, this is not the case and we argue the hetereogeneity in karst architecture primarily controls the spatial and temporal variation in the presence of the daily oscillations, however, hydraulic connectivity does not cause the phenomenon. Thus, we argue that it would not be appropriate to include hydraulic connectivity here as suggested because this paragraph discusses possible drivers of the daily oscillations.

-Can you quantify the T amplitude - strenght of cpd signal relationship?

We thank the reviewer for this comment. We found that the T amplitude-strength of cpd signal relationship was quite complex, which we argue is further evidence that this is a biological process rather than a physical one. Unfortunately, we could not quantify this relationship but argue, nonetheless, that it is an interesting outcome from this study. We will reword the text to ensure a statistical relationship is not alluded to.

-How strong is the relationship? Is it significant?

The relationship between 2-day moving average of air temperature and drip rate during the period 1/2/14- 19/2/14 is significant and weak ($\tau$= -0.21, significant at 95%). We will amend the text to reflect this information.

-Can you quantify the negative relationship between cloud cover and strength of cpd signal?

Thank you for this query. We do not use cloud cover data, rather we use daily ET to examine the relationship between radiation and cpd signal on a multi-day timescale and air temperature as a proxy for ET on a sub-daily timescale. We would like to direct the reviewer to our response above regarding the T amplitude-strength of cpd signal relationship.

-This subsection is quite long, moving this statement to the implications in the next subsection could shorten it

Thank you for this suggestion. We argue that this paragraph is needed here so there is a logical flow from discussion of solar radiation to evapotranspiration. However, we agree that this section is too long and will insert a further subsection '4.2.5.1 Scenarios for solar driven daily cycles of phreatophytic evapotranspiration' to reduce the length of this section.

-I really like the conceptual elaborations in Fig 6. They could be improved when adding small graphs of expected daily variation of drip rates in the 9 Figures (similar to Fig 7 in Barbera & Andreo, 2011)

Barberá, J.A., Andreo, B., 2011. Functioning of a karst aquifer from S Spain under highly variable climate conditions, deduced from hydrochemical records. Environ. Earth Sci. 65, 2337–2349. doi:10.1007/s12665-011-1382-4

Thank you for this suggestion, we had similar graphs on the original sketch and we will add them into the Fig 6.

-The outcomes of this study are clearly relevant for paleoclimte reconstructions by speleothems, as well expleined in this subsection. But the implications are also of highest relevance for karst modeling and karst water resources assessment.

There are still no approaches that do consider the impacts of vegetation on recharge
dynamics, neither in process-based karst models (Hartmann et al., 2014, 2015) nor in
empirical recharge estimation approaches (Allocca et al., 2014; Andreo et al., Andreo et al.,
2008)

Allocca, V., Manna, F., De Vita, P., 2014. Estimating annual groundwater recharge coefficient
for karst aquifers of the southern Apennines (Italy). Hydrol. Earth Syst. Sci. 18, 803–817.
doi:10.5194/hess-18-803-2014

Andreo, B., Vías, J., Durán, J., Jiménez, P., López-Geta, J., Carrasco, F., 2008. Methodology
for groundwater recharge assessment in carbonate aquifers: application to pilot sites in
southern Spain. Hydrogeol. J. 16, 911–925.

Hartmann, A., Gleeson, T., Rosolem, R., Pianosi, F., Wada, Y., Wagener, T., 2015. A large-
scale simulation model to assess karstic groundwater recharge over Europe and the
Mediterranean. Geosci. Model Dev. 8, 1729–1746. doi:10.5194/gmd-8-1729-2015

Hartmann, A., Mudarra, M., Andreo, B., Marín, A., Wagener, T., Lange, J., 2014. Modeling
spatiotemporal impacts of hydroclimatic extremes on groundwater recharge at a
Mediterranean karst aquifer. Water Resour. Res. 50, 6507–6521.
doi:10.1002/2014WR015685

We thank the reviewer for their comments; we would like to refer them to "Specific
Comment #4" which we believe addresses their queries.

---

## Author Response (AR1)

Material Science and Engineering Building

University of New South Wales

Kensington

Sydney

GFZ German Research Centre for Geosciences

Section 5.4 Hydrology

Germany

Dear Dr Blume,

I am writing on behalf of the authors of the manscript titled "Solar forced diurnal regulation of cave drip rates via phreatophyte evapotranspiration" to thank you for your time in editing our manuscript further and for your insightful suggestions. We have responded to the recommendations point by point in the document attached and we have submitted the revised manuscript as requested. If you require any further information, please do not hesitate to contact me.

Warm regards,

Katie Coleborn

**Authors' response to editor's comments**

Dear Authors, while the referees both see the scientific significance of this manuscript, they also both find that the scientific quality can be improved and recommend reconsideration of the manuscript after major revisions.
Please revise your manuscript according to their comments and suggestions.

Additionally, a few more detailed recommendations:
As both referees agree on the fact that the description of the methodology should be improved I also recommend adding at sketch, as suggested by referee #2.

We have again revised the methodology, and have now included a sketch showing the steps that were used to map and identify frequency components in the drip discharge rate, temperature and barometric pressure data. This also includes an improved outline of the advantages of synchrosqueezing over traditional signal processing methods (e.g., Fourier and wavelet transform). We hope that this satisfies both reviewers' and the editors requests.

Referee #1 asks whether or not the measurements of drip rates are representative, please provide a short discussion of this.

Thank you for raising this query. We will include a more detailed site description to address this issue:

"The drip sites were chosen to be representative of the cave location. We used a stratified sampling method where a transect of the cave was used to select three locations (G, M and LR) that satisfied the following criteria 1) there were actively dripping speleothems, 2) spatially distant from the other locations and 3) different depths within the cave. Individual drips were sampled randomly at each location, with selection guided by practical constraints such as the stalagmite surface being suitable for placement of a logger and the drip falling from high enough to activate pressure sensor on logger"

Referee #1 suggests showing higher resolved time series of ET and I agree that it would be helpful to show hourly or even higher resolved values instead of daily values here.

Thank you for your comment. We appreciate the benefit of using higher resolution ET data, however, we have used the highest resolution ET data available to us. We have used air temperature data at 15 minute intervals and have demonstrated the strong relationship between ET and temperature in the manuscript.

Referee #1 asks for an explanation of the term "negative hydraulic lift" and I am also thinking that you mistakenly mixed the two terms "hydraulic lift", which normally refers to movement of water from the roots into the soil, and "negative sapflow". Please check and clarify.

Thank you for bringing this point to our attention, we have amended the text in the following way:

"Burgess et al (2001) measured sap flow in Eucalypt tap roots, finding tap root sap flow peaked around 1 pm and negative sap flow values indicated reverse (acropetal) flow between 7pm- 7am."

From your responses to the specific comments #1 and #4 of referee #2 it is not clear how you intend to revise your manuscript.

In regards to the response to comment #1 (reviewer #2) we have added a few sentences to the introduction addressing the relevance of this manuscript along the lines of:

"This study has important implications for understanding karst unsaturated flow processes and karstic groundwater recharge. Currently, most karst models use very simplistic representations of unsaturated flow, if it is considered at all (Hartmann et al., 2014a). This study highlights the importance of vegetation dynamics on vadose flow and recharge making it significant to karst modelling research and speleothem-based paleoclimate studies which focus on the impact of vegetation dynamics on proxy records (Treble et al., 2015, accepted for publication 8/4/16.)"

Please add the recommended points to the discussion as suggested.

In response to comment #4 (reviewer #2) we have added the following lines to the discussion:

"This study clearly demonstrates the potential for vegetation to impact karst water recharge making this research relevant to karst modelling and karst water resources assessment. Currently, there are no approaches that consider the impacts of vegetation on recharge dynamics in process-based karst models (Hartmann et al., 2014b, 2015) or in empirical recharge estimation approaches (Allocca et al., 2014; Andreo et al., 2006)."

Please also add more quantification of the identified relationships and their significance to section 4.2.5. Solar driven daily cycles of vegetative (phreatophytic) evapotranspiration.

In response to the reviewer comments we have quantified the relationship between air temperature and drip rate in section 4.2.5. and have included the statistical outcome in the revised manuscript. We have provided an explanation as to why the relationship between T-amplitude/cloud cover and strength of cpd signal cannot be quantified.

And again I would like to come back to my original suggestion of a slightly more detailed time lag analysis applying ccf or similar to all the periods with diurnal oscillations - comparing drip rates either with the temperature time series or with a time series of hourly (or 15 minute) ET. Then you would obtain a lag time and a correlation value for each of these periods (this could be presented in a plot or a table) and it would be possible to see if these values are dependent for example on the season or if they are mainly site specific.

Thank you for this suggestion, we have performed the cross correlation analysis for
temperature and drip rate for the individual time periods where the oscillations in drip rate
occur to explore how season and site explain the amount of variance in the timing of
minimum drip rate. We have added Table 2 and updated the text to reflect the use of this
quantitative approach:

[revised manuscript text omitted]

---

## Author Response (AR2)

Material Science and Engineering Building

University of New South Wales

Kensington

Sydney

GFZ German Research Centre for Geosciences

Section 5.4 Hydrology

Germany

Dear Dr Blume,

I am writing on behalf of the authors of the manuscript titled "Solar forced diurnal regulation of cave drip rates via phreatophyte evapotranspiration" to thank you and the two reviewers for providing insightful and detailed comments on our manuscript. We have taken these suggestions into account in the revised manuscript and have provided a point-by-point response. We hope that our amendments and clarifications are satisfactory. Please do not hesitate to contact me if you require any more information.

Warm regards,

Katie Coleborn

**Response to manuscript review**

Dear Authors,

I agree with the referees that you have improved your manuscript significantly and I am positive that we will be able to publish it as soon as you made a few minor revisions as suggested by the referees and myself. Please find my suggestions below.

Best regards,

Theresa Blume

Lag time analysis: could the lag also be >24 hours? Please discuss and check. Did you try shifting the time series for > 24 hours, i.e. several days? Are the results the same?

Lags greater than 24 hours were investigated, and in some cases the autocorrelations were slightly stronger than observed in the 0-24 hour period for example, between 11/02/2013-21/03/2013 at site G1 the strongest correlation occurred at 2.5 days (-0.86). However, this is to be expected in a time series whose fundamental frequency is 12 or 24 hrs. Where stronger autocorrelations were observed >24 hrs, they were typically harmonics of the fundamental frequency occurring at 12 hour intervals, and therefore not a new signal.

Please indicate if the correlations in table 2 are significant (by adding a star or putting them in bold letters), or if they are all significant say so in the caption. I would keep the table in its current detail and not cumulate the values to averages as suggested by Reviewer 2.

Thank you for this suggestion, we have updated the table to specify all correlations are significant.

I would call Figure 2 a table, not a figure. I am not sure if this was meant by the reviewer when asking for a sketch. I think that sketch was supposed to help explain to people who aren't experts in signal processing the steps happening when running the analysis with the toolbox. Would it be possible to provide something like that? If it is not possible to provide such a sketch, very briefly state the different steps within the SST analysis (1 or 2 sentences). If I understand correctly (I am not expert) this is a "sharpening" of a wavelet transform – so to point out the advantage over the wavelet transform it might also be helpful to put those two plots next to each other. That way people can directly see how the CWT "transforms" into the SST and how much more helpful the SST is. I agree with the reviewer that in its current state Figure 2 is basically just text and could be also moved to the text (most of it is in the text already anyway). Also, some parts of this table are not really helpful in understanding the methods (e.g. step 1 or the remark about the color scale).

Thanks for your detailed comments. As requested by reviewer #1, we have now merged Figure 2 into the methodology (excluding the unhelpful bits).

Note that synchrosqueezing was officially added to MATLAB as of release R2016a, where it was named the "wavelet synchrosqueezed transform" (WSST). We have added this to the text, and have revised our terminology and abbreviations throughout the manuscript in order to avoid confusion.

Thanks also for your suggestion of to illustrating the "sharpening" of the time-frequency
content by comparing results from the new technique (the wavelet synchrosqueezed
transform - WSST) with traditional wavelet analysis (continuous wavelet transform - CWT).
We have made a new figure showing the time-frequency content obtained by applying both
techniques side-by-side using an example. We believe this new figure is very useful and
appropriately illustrates the advantages of WSST in the context of our manuscript.

The description of the different steps involved in WSST is complicated, as this is an advanced
signal processing method. Now that WSST has become a recognised and readily usable
technique (e.g. through MATLAB), we do not think it is necessary to elaborate on the details
as we merely apply this technique to reveal phreatophyte evapotranspiration in cave drip
discharge. Instead, we refer the technically minded and interested reader to the literature
which contains the details.

Figure 6: remove 96pt from the legend or explain it if you find it necessary. "Daily moving
average" might be more directly understood.

Thank you for this comment, we have made the changes to Fig 6.
Figure 7:add unit (years) to the last three plots. Is this supposed to be the age of the trees?
Then maybe change the zero to 1 and write "age: 1 year" into the figure.
Figure 7: does it mean something different if the root is accessing the blue part of the
storage or just the grey part? If yes please explain, if not it might be helpful to make this
similar in the different sketches.
Figure 7: do you really mean hydraulic lift? Or just root water uptake? If you just meant root
water uptake do not call it hydraulic lift – see my earlier remarks about this.
Figure 7: sometimes the blue area fills a larger fraction of the box than in other plots (c and
d) – does this mean the storages are filled to a different extent? If not, please keep it
consistent.
Figure 7: I would add a key word as a header to the different sketches and maybe arrows to
show which ones you are supposed to compare. Otherwise you automatically start
comparing a) and c) because they are right next to each other, this can cause confusion.
Examples for such headers: short flow path, long flow path, large storage, small storage, low
uptake, high uptake,…

Thank you for your suggestions on ways to improve Fig 7, we have taken them all into
account in the revised figure.

l.525: could this also be the result of compensation as the preferred storage providing water
for root uptake dries out and the tree has to shift uptake to wherever water is still available
(probably at greater depth)? This would mean the roots were already present at this depth
and were put to use once it became necessary. Can we really attribute this strong change
between May and August solely to root growth? Assuming a linear relationship this would
mean a shortening of the flow path to 1/3 of the original length so an increase in rooting
depth by a factor of 3. I know this is just a very rough calculation, but please make sure if
this is really the only and most probably hypothesis.

We have amended the manuscript to indicate that our explanation for why the lag time
changes is speculative. We have also amended the text to include the alternative scenario
suggested.

l.121: what does "well decorated" mean? Please rephrase or explain.

Well decorated means that the cave has an abundance of speleothems, we have amended
the text to clarify this term.

**Reviewer #1**

Summary

The authors implemented many comments raised by the reviewers and the editor and
improved the manuscript.

While this is good news, I still stumbled over several issues which need to be resolved
before the manuscript can be accepted.

- tendency for overstatements: At various places the authors state that

L29: "This is the first observation of tree water use … "

L497: The first study that shows that tree water use affecting cave drip water …"

L 639: "This is the first volumentric observation if tree water use in cave drip water"

Given the measurements they show that there are diurnal fluctuation in cave drip rates.
That this is due to tree water use is likely but not directly measured!

We thank the reviewer for raising this issue and have amended the text to better reflect the
nature of our findings for example:

"L29: This is the first indirect observation of tree water use in cave drip water…"

- discussion: Figure 1 shows that the drip rates of sites G6, G8, G12 and M1 are very low
compared to other sites. These low values are likely to be affected by considerable
measurement uncertainty. Some of the mentioned sites even show fluctuations typical for
being at the low end of sensor resolution.

Despite of this potential uncertainty, very specific results of single sites are discussed in
terms of tree root dynamics (L560-564) or storage volume (L570-579). I strongly recommend
to cut these paragraphs and unclear speculations. This would also help to shorten the
somewhat lengthy discussion.

We have included specific examples of periods of fluctuation that relate to the different
scenarios in Fig.7 at the request of the editor in a previous iteration of the manuscript. We
agree with the reviewer that the measurement uncertainty could be high for sites such as
M1, however, we do not think this detracts from the overall argument as we are not stating
that transpiration is using X volume of cave drip water, rather we refer to the drip site flow
type in general and infer the karst architecture and how this influences the drip water
response to tree water use.

- conceptual diagram: I really appreciate addition of the time series sketches. Why are there
no differences in a and b despite flow path length? Why is there no difference in h) and i)?
Please increase font size.

We have increased the font size to highlight the differences between the sub figures.

- Figure 2 and illustration of spectral method. Both reviewers where asking for more details
of the method. Now please put the text of Fig.2 into the main text, excluding some of the
unimportant info, e.g. customising color scale etc. If possible give the equation for F f,t

We have now merged the text from Figure 2 into the methodology, as requested. We do not
think that the reader would gain from re-stating what is involved in arriving at F(f,t) because
it is complicated (not a simple equation) and has been documented in peer-reviewed
literature that is properly referenced in our manuscript.

L205: why is the identification subjective? Why not use s threshold of the normalised
amplitude? This would make the results more objective, especially if it is argued later on to
establish a protocol for drip rate measurement and analysis.

We thank the reviewer for this comment. The frequency components can be distinguished
easily from the chaos in the pseudo-colour plots. We have further explained how the criteria
for strength, continuity and stability can be automated if the user chooses to do so.

- Table 2 and crosscorrelation results: I also believe that this standard analysis is valuable.
Please connect these results with the SST results for the given periods, such as cycles per
day and amplitude / phase. This would help to establish the SST method. Also add the
minimum / maximum times info given in the text at L279 - 335 as extra column to the table.

We thank the reviewer for appreciating the value in the analysis we have included. We have
included additional columns for the timing of the max and min drip rate and specified the
periods with 2 cpd fluctuations.

- correlation between potential evaporation and temperature / radiation (L353, L475) is by
definition because it is computed from both variables! Please cut these sentences.

Thank you for this suggestion, we have removed the sentences.

- Fig. 6 and related text L464-468: I think that the negative correlation between the 2day
averaged Tair and drip rates could be coincidence and not a regular causality as it seems
that this only occurred at one site in one week? To establish a better causality would need
much more careful investigation and analysis. As this is not the focus of the manuscript I
would recommend to cut this part.

We thank the reviewer for this comment, we believe that there is a valid causality between
2 day averaged air temperature and drip rate and have only used one example in the text
for brevity and to support the link between the drip rate and tree water use.

- link between diurnal temperature range and 1cpd drip rate signal, L457-463:

Given that the first 10 days in Feb. show high temperature ranges and ET rates but no
diurnal cycle in drip rates indicates that more than high transpiration rates are required, to
lead to a diurnal fluctuation in drip rates. Considering a simple leaky bucket, it requires that
the storage volume, inflow, root water uptake and outflow need to reach a certain state to
result into a fluctuating outflow.

We agree with the reviewer that there are multiple factors influencing the occurrence of the
diurnal fluctuations. In this section we highlight the influence of solar radiation and in later
sections of the discussion, take into consideration the influence of other factors such as
storage volume, root uptake, flow path length etc. We highlight the point that complexity of
the karst architecture in determining how the diurnal fluctuations are exhibited in regards
to lag time and seasonal timing. For example, "Line 641-643 We proposed that the
complexity of flow pathways in the karst system accounted for the spatial and temporal
variation in the daily fluctuations of drip rate."

L90-96 This paragraph sounds like final conclusions.

We thank the reviewer for this comment. We included these details at the request of the
previous reviewer who suggested emphasising the wider implications of this study in the
introduction.

Table1: means of total flow volume are reported? Does this means that there are several
drip counters per site which have been averaged? Or are these seasonal averages across
different years? Please check this.

The mean total flow volume refers to the monthly mean at each site. We have amended the
caption to reflect this.

L155: check unit of radiation

Thank you for raising this issue, we have corrected the units.

L303-308: somewhat loose paragraph

This paragraph has been reworded more concisely.

**Reviewer #2**

In their re-submission, Coleborn et al present a significantly improved versuion of their
previously submitted manuscript. The introduction now includes some statement about the
relevance of this study concerning karst modeling. A very nice sketch to elaborate the
approach was added and the methodology chapter was improved. Also the discussion is
more complete. A time lag analysis with temprature and drip rates was added.

Overall, I feel confident recommending the manuscript for publication now.

We thank the reviewer for their comments.

If the authors think this will help, they may replace the header "comment" with the
"elaborations" in the new figure 2 (column 2).
We have included Figure 2 into the main text, as requested by reviewer #1 and the editor.

Also, in order to shorten Table 2 and still make their point, they may consider providing the
mean lag time and correlation coefficients (with their standard deviations) for each location.

At the request of the editor we have preserved the format of Table 2.

---

## Author Response (AR3)

Material Science and Engineering Building

University of New South Wales

Kensington

Sydney

GFZ German Research Centre for Geosciences

Section 5.4 Hydrology

Germany

Dear Dr Blume,

I am writing on behalf of the authors of the manuscript titled "Solar forced diurnal regulation of cave drip rates via phreatophyte evapotranspiration". Thank you for your suggestions on how to further improve the manuscript. I have included the tracked changes of the manuscript below in lieu of a point-by-point response and have uploaded a clean version of the manuscript separately. Please do not hesitate to contact me if you require anything further.

Warm regards,

Katie Coleborn

[revised manuscript text omitted]

Comment [GCR2]: Theresa: not the other way round?

Comment [GCR3]: Theresa: Is this text or caption? If it is caption, delete the "shows" and replace by ":"

Also explain what the negative/positive time lags mean. Does a positive time lag of 2 hours mean that drip rate lags behind temperature by 2 hours?

Comment [GCR4]: Theresa: combine the columns below max and min drip rates into one each giving the time range as "9:00-12:00". It might also be helpful to use the 24 hour clock.

[revised manuscript text omitted]

---

## Author Response (AR4)

324 Material Science and Engineering Building

University of New South Wales

Kensington

Sydney

2052

GFZ German Research Centre for Geosciences

Section 5.4 Hydrology

Germany

Dear Dr Blume,

I am writing on behalf of the authors of the manuscript titled "Solar forced diurnal regulation of cave drip rates via phreatophyte evapotranspiration" to thank you for the most recent comments, we have made the suggested changes to the sentence in the abstract and adjusted the figure captions.

Warm regards,

Katie Coleborn